# MTIR-SQL: Multi-turn Tool-Integrated Reasoning Reinforcement Learning for Text-to-SQL

## Abstract

As large language models (LLMs) are increasingly used in Text-to-SQL tasks, Reinforcement Learning (RL) has become a common method for improving performance. Existing methods primarily rely on static execution feedback, which restricts real-time error correction. However, integrating multi-turn tool invocation along with dynamic feedback could significantly improve adaptability and robustness, ultimately enhancing model performance. To address these issues, we propose **MTIR-SQL, an innovative Multi-turn Tool-Integrated Reasoning reinforcement learning framework for Text-to-SQL**. Our approach introduces an execution-aware multi-turn reasoning paradigm that seamlessly incorporates database execution feedback at each reasoning step, enabling context-sensitive query generation and progressive refinement throughout the reasoning process. The framework extends the GRPO algorithm to accommodate complex multi-turn interaction scenarios. Considering the training instability characteristics of MTIR and the potential for significant Deviation of model distribution from the initial model, we enhance the GRPO algorithm by adding a trajectory filtering mechanism and removing KL loss constraints. Experimental results demonstrate that MTIR-SQL, with 4B parameters, achieves 64.4% accuracy in the BIRD Dev and 84.6% execution accuracy in the SPIDER Dev, significantly outperforming existing approaches.

## 1 Introduction

Text-to-SQL, the task of automatically translating natural language questions into executable SQL queries, is a key technique for lowering the barrier to database access (Liu et al., 2025). By enabling non-technical users to query structured data in natural language, it has found wide applications in business intelligence, data analytics, and interactive question answering (Hong et al., 2025).

Existing approaches to Text-to-SQL generally fall into three paradigms: (i) supervised fine-tuning (SFT) of domain-specific open-source models (Li et al., 2025b; 2024b); (ii) prompting closed-source large language models (LLMs) with advanced reasoning strategies such as chain-of-thought (Li et al., 2025a; Zhai et al., 2025; Pourreza et al., 2024); and (iii) reinforcement learning (RL)-based methods that optimize model reasoning with algorithms such as PPO and GRPO (Pourreza et al., 2025; Ma et al., 2025; Yao et al., 2025; Dai et al., 2025). In particular, RL methods leverage final SQL execution results as reward signals for policy optimization. However, current practices treat execution feedback merely as scalar rewards, wasting rich tool information and leaving static LLMs unable to adapt their reasoning dynamically.

Recently, Multi-turn Tool-Integrated Reasoning (MTIR) has emerged as a promising paradigm for enhancing the reasoning capabilities of LLMs. By interleaving model reasoning with external tools—such as search engines, Python interpreters, and SQL executors—MTIR enables LLMs to overcome intrinsic limitations in computation, retrieval, and structured manipulation. Several studies have demonstrated the benefits of RL-based optimization in tool use: Search-R1 (Jin et al., 2025) explores dynamic tool invocation for question answering, while ToRL (Li et al., 2025c) and Effective CIR (Bai et al., 2025) design tailored RL recipes for mathematical reasoning. More recently,

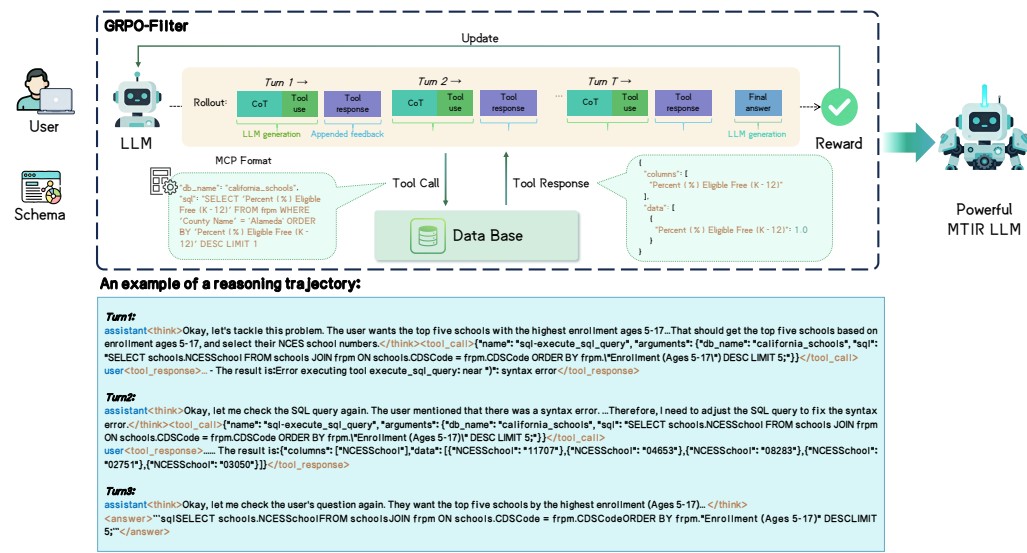

Figure 1: Overview of the MTIR-SQL framework. The framework integrates multi-turn reasoning with execution feedback and extends GRPO with trajectory filtering to enable dynamic correction and stable training, thereby enhancing SQL generation accuracy in complex scenarios.

the ReEx-SQL (Dai et al., 2025) framework extends TIR to Text-to-SQL, providing early evidence of its potential in structured query tasks.

Despite this progress, three fundamental challenges remain. On the **tool side**, SQL-oriented MTIR remains underexplored; existing efforts often rely on ad-hoc token mechanisms for tool invocation, limiting interoperability, extensibility, and compatibility with diverse database operations (Jin et al., 2025; Dai et al., 2025). On the **RL side**, dominant methods such as GRPO suffer from reward collapse and difficulty in modeling long-horizon dependencies, leading to instability in multi-turn tool interaction. On the **framework side**, current Text-to-SQL studies adopt heterogeneous, non-standardized implementations that lack modularity and generality (Dai et al., 2025; Ma et al., 2025; Yao et al., 2025; Gajjar et al., 2025).

To address these issues, we propose **MTIR-SQL**, a reinforcement learning framework for Multi-turn Tool-Integrated Reasoning in Text-to-SQL (Figure 1). MTIR-SQL extends GRPO to handle complex multi-turn interactions and introduces two key modifications: (i) a trajectory filtering mechanism to discard invalid rollouts and (ii) the removal of KL regularization to mitigate distributional collapse during training. Built on top of RL-Factory (Chai et al., 2025) with standardized MCP-compatible tool invocation, our framework ensures extensibility and interoperability.

Our contributions are summarized as follows:

- **MTIR-SQL Framework.** We introduce a novel RL framework for Text-to-SQL that enables LLMs to reason interactively and directly optimize via SQL execution feedback. It incorporates tool response tokens masking for stable training and supports multi-turn iterative reasoning and execution verification.

- **GRPO Extensions.** We extend GRPO with SQL execution rollout expansion and trajectory filtering to stabilize training in multi-turn tool-use scenarios, effectively mitigating reward collapse.

- **Strong Empirical Results.** On the BIRD dataset, MTIR-SQL trained on Qwen-3-4B achieves a 16% absolute improvement over baselines, matching the performance of recent 7B-coder models. It attains competitive execution accuracy, demonstrating its robustness and effectiveness.

## 2 RELATED WORK

### 2.1 RL FOR TOOL-INTEGRATED REASONING

Tool-Integrated Reasoning (TIR) has emerged as a key paradigm for augmenting large language models by enabling interaction with external tools and APIs (Zhang et al., 2025). Early work focused on single-turn tool invocation through supervised fine-tuning approaches, demonstrating effectiveness across domains, including mathematical reasoning, code generation (Mai et al., 2025), and search integration (Jin et al., 2025). Multi-turn TIR enables iterative reasoning through sequential tool interactions, where models repeatedly generate tool calls, execute it, and refine based on results (Mai et al., 2025; Shang et al., 2025; Wang et al., 2025; Zeng et al., 2025; Zhao et al., 2025). However, training stability remains a critical challenge due to distribution drift from external tool outputs and error accumulation across reasoning rounds, often leading to training instability and entropy collapse. Recent work like SimpleTIR (Xue et al., 2025) addresses these issues by filtering empty rounds in multi-turn reasoning, achieving state-of-the-art performance on mathematical tasks. Despite these advances, optimizing Multi-turn TIR for complex tasks remains challenging (Lin & Xu, 2025; Dong et al., 2025a;b; Yu et al., 2025). We apply recent MTIR advancements, including filtering and handling invalid turns, to the Text-to-SQL domain, improving execution feedback management, multi-table relationship handling, and ensuring SQL semantic correctness across iterations.

### 2.2 TEXT-TO-SQL

Text-to-SQL aims to automatically convert natural language questions into executable SQL query statements, enabling natural language interfaces for databases. The field has evolved through three main paradigms: supervised fine-tuning methods that train specialized models on domain-specific datasets (Li et al., 2024b; 2025b; Yang et al., 2024; Qin et al., 2025), using closed-source large models with prompt engineering and chain-of-thought reasoning to handle complex multi-table joins and nested queries (Li et al., 2025a; Zhai et al., 2025; Pourreza et al., 2024; 2025; Lyu et al., 2025; Pourreza & Rafiei, 2023; Xie et al., 2024; Cao et al., 2024), and reinforcement learning approaches using algorithms such as GRPO to enhance reasoning capabilities and generalization (Dai et al., 2025). Despite recent advancements, current RL-based methods exhibit significant limitations. They rely on static context during generation and lack mechanisms for validating or correcting intermediate reasoning steps, resulting in errors that cannot be self-corrected (Ma et al., 2025; Yao et al., 2025; Gajjar et al., 2025). Execution feedback is treated as a reward signal rather than dynamically integrated, hindering the model's ability to adapt to complex scenarios. The challenge persists in incorporating execution feedback while managing database results, multi-table relationships, and ensuring SQL semantic correctness. To address these issues, we introduce Multi-turn Tool-Integrated Reasoning in the Text-to-SQL domain, enabling the model to improve performance through iterative use of external tools.

## 3 METHODOLOGY

We propose an SQL-integrated reinforcement learning framework with GRPO-Filter, which combines unconstrained optimization, selective rollout filtering, and multi-turn reasoning to improve decision-making. The model dynamically interacts with SQL execution, refining its output through iterative feedback. A reward mechanism focused on format, execution, and result correctness guides the generation of high-quality SQL queries.

### 3.1 SQL-INTEGRATED RL WITH GRPO-FILTER

We formulate the reinforcement learning framework with SQL execution tool $\mathcal{E}$ as follows:

$$\max_{\pi_\theta} \mathbb{E}_{x \sim D, y \sim \pi_\theta(\cdot|x;\mathcal{E})}[r_\phi(x, y)], \tag{1}$$

where $\pi_\theta$ is the LLM policy and $r_\phi$ is the reward function. Unlike prior reinforcement learning methods that primarily rely on the policy LLM $\pi_\theta(\cdot|x)$ to generate rollout sequences, our framework explicitly incorporates SQL execution-guided reasoning via $\pi_\theta(\cdot|x;\mathcal{E})$, which can be formulated as

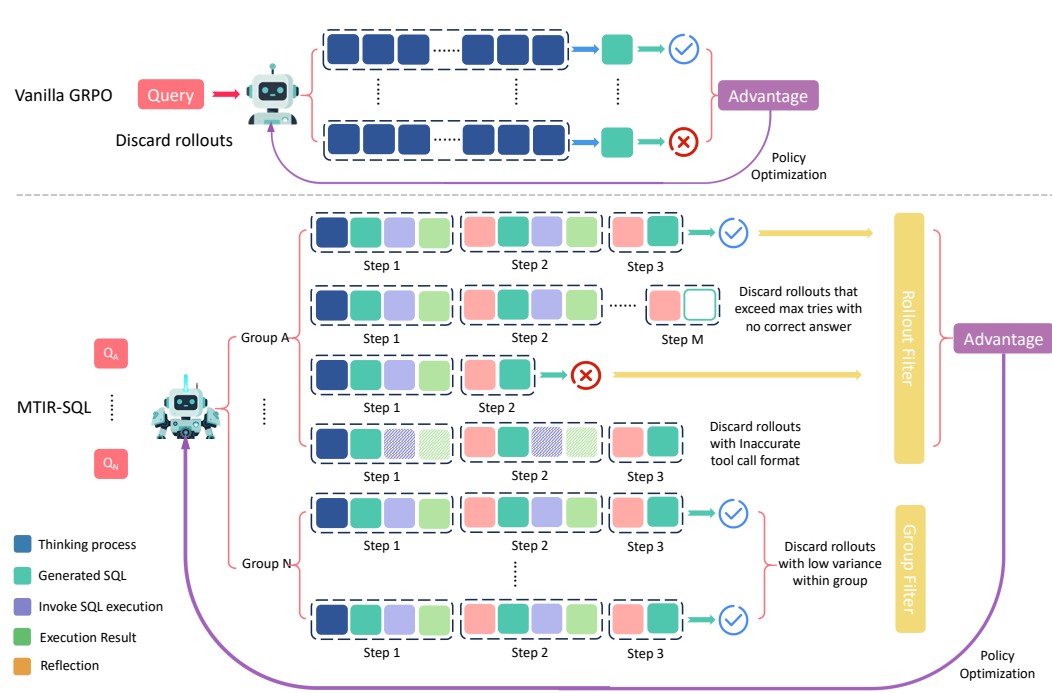

Figure 2: Compared to vanilla GRPO, our framework removes the KL constraint, introduces quality-aware rollout filtering, and extends to multi-turn reasoning with SQL execution feedback for more stable and accurate policy optimization.

$\pi_\theta(\cdot|x) \otimes \mathcal{E}$, where $\otimes$ denotes the interleaved SQL generation and execution feedback. This enables more effective decision-making in SQL generation tasks by leveraging real-time execution results to guide the model's reasoning process.

Our approach introduces **GRPO-Filter**, an enhanced variant of Group Relative Policy Optimization (GRPO) specifically designed for complex multi-turn interactive scenarios. GRPO-Filter incorporates three key innovations:

**Unconstrained Optimization:** Unlike standard GRPO, we remove the KL divergence constraint between the policy and reference model, allowing for more flexible policy updates:

$$\mathcal{L}_{\text{GRPO-Filter}} = -\mathbb{E}_{(x,y)\sim\mathcal{D}} \left[ \frac{\pi_\theta(y|x; R)}{\pi_{\text{ref}}(y|x; R)} \cdot A(x, y) \right], \tag{2}$$

where $A(x, y)$ represents the advantage function, eliminating the traditional KL penalty term $\beta \cdot \text{KL}(\pi_\theta || \pi_{ref})$.

**Selective Rollout Filtering:** To ensure the stability of gradient estimation and the efficiency of policy learning, we implement a multi-dimensional quality-aware filtering mechanism. Let $G_x = \{y_1, y_2, \dots, y_K\}$ denote a group of $K$ trajectories generated for input $x$. The set of retained trajectories is defined as:

$$\mathcal{T}_{\text{filtered}} = \{(x, y) \in \mathcal{T}_{\text{rollout}} : \mathcal{F}(x, y, G_x) > \tau\}, \tag{3}$$

where $\tau$ serves as the binary acceptance threshold (set to 0 for strict boolean filtering). The filtering function $\mathcal{F}(x, y, G_x)$ is a composite indicator designed to filter out noise and uninformative samples. It is formally defined as the conjunction of two criteria:

$$\mathcal{F}(x, y, G_x) = \mathbb{I}_{\text{valid}}(y) \cdot \mathbb{I}_{\text{div}}(G_x). \tag{4}$$

Specifically, these components address the following aspects:

- *Execution Validity* ($\mathbb{I}_{\text{valid}}$): This term filters out invalid interaction patterns, specifically defined as trajectories where the model performs tool invocations for more than two turns without yielding a final answer, or encounters fatal execution errors. This ensures that the policy prioritizes efficient and conclusive reasoning paths.

- *Representation Diversity* ($\mathbb{I}_{\text{div}}$): To prevent mode collapse and ensure meaningful advantage computation, we discard groups with insufficient variance. Specifically, $\mathbb{I}_{\text{div}}(G_x) = 1$ if the standard deviation of rewards within group $G_x$, denoted as $\sigma_R(G_x)$, exceeds a minimum threshold $\epsilon_\sigma$.

**Multi-turn Extension:** GRPO-Filter extends the original framework to handle complex multi-turn interactions by maintaining conversation context and enabling iterative reasoning across multiple dialogue turns:

$$\pi_\theta(y_t|x, h_{<t}; R) = \pi_\theta(y_t|\text{concat}(x, h_{<t}); R), \tag{5}$$

where $h_{<t}$ represents the conversation history up to turn $t$, and $y_t$ is the response at turn $t$.

This multifaceted approach allows GRPO-Filter to effectively optimize policies for reasoning-intensive tasks while maintaining training stability and improving sample efficiency through selective learning from high-quality experiences.

## 3.2 Interaction with SQL Execution Environment

The integration of SQL and its execution interface with large language models (LLMs), which are capable of comprehending and generating query intentions, can significantly enhance the automation of complex database operations. In an LLM-based SQL tool invocation environment, the system should exhibit human-like interactive and reasoning behaviors. These behaviors include generating syntactically correct and logically sound SQL queries from natural language questions, invoking database execution interfaces at appropriate moments, and executing queries safely. Additionally, the system should carefully interpret query results, verify their correctness, and refine subsequent problem decomposition or query generation strategies based on feedback. This capability is cultivated through guiding the model via multi-turn interaction and reflective learning with the SQL execution environment. Detailed prompt of sql Execution can be found in Appendix D.1.

With the support of SQL tools, the model dynamically incorporates database query results into the reasoning process through multi-turn execution, as illustrated in Figure 1. Unlike conventional methods that generate a complete SQL query until an end-of-sequence (EOS) token is produced, our approach constructs the full reasoning trajectory through continuous interaction with the SQL execution environment. The first interaction begins with a system prompt followed by the user's question, with detailed content available in Appendix D.1. The model, acting as the assistant, generates an initial response until it outputs an EOS token. If no SQL tool call is detected, the process terminates. When an SQL query is identified, the environment service extracts and safely executes it, then appends the execution result to the dialogue context in the user role. The model subsequently continues its reasoning as the assistant based on the updated context, producing the next turn of response. This multi-turn process iterates until the model returns a final answer or a maximum number of turns, denoted as $T$, is reached. Detailed content can be found in Appendix B.

## 3.3 Reward Design

To optimize policy effectively, we introduce a streamlined reward mechanism that focuses on critical elements of SQL query quality. This framework incorporates three key factors—syntax validity, execution feasibility, and semantic precision—each providing distinct guidance to ensure the model generates SQL queries that are syntactically correct, executable, and semantically meaningful.

**Format Reward.** We guide the model to maintain a specific sequence of tags, ensuring a structured response. The response should follow a strict order: starting with `<think>`...`</think>`, followed optionally by `<tool_call>`...`</tool_response>`, and concluding with the `<answer>`...`</answer>` tag. Additionally, all tools must be used within `<tool_call>`...`</tool_call>` and `<tool_response>`...`</tool_response>` tags to maintain a standardized flow.

$$R_f = \begin{cases} 0.1, & \text{if the format is correct,} \\ -0.1, & \text{if the format is incorrect.} \end{cases} \quad (6)$$

**Execution Reward.** This reward evaluates the syntactic correctness and executability of the generated SQL. It prevents the model from producing invalid or overly complex queries. If the SQL statement fails to execute, the model will not receive subsequent rewards. Furthermore, the execution time is constrained to discourage the generation of unnecessarily complex queries:

$$R_e = \begin{cases} 0.1, & \text{if the SQL query is executable,} \\ 0, & \text{if the format is incorrect,} \\ -0.1, & \text{if the SQL query is not executable.} \end{cases} \quad (7)$$

**Result Reward.** The correctness of query results is a crucial measure of semantic fidelity. To encourage faithful reasoning, we design the result reward to strongly differentiate between correct and incorrect outputs:

$$R_r = \begin{cases} 1, & \text{if the query result is correct,} \\ 0, & \text{if the query result is not correct.} \end{cases} \quad (8)$$

## 4 EXPERIMENTS

### 4.1 EXPERIMENTAL SETUP

**Datasets.** We train and evaluate our model on two Text-to-SQL benchmarks, SPIDER (Yu et al., 2019) and BIRD (Li et al., 2023), which assess different aspects of the task. SPIDER is a large-scale, cross-domain benchmark focused on SQL complexity, with 10,181 questions and 5,693 unique queries across 200 databases. BIRD addresses real-world scenarios, featuring 12,751 question-SQL pairs on 95 large-scale databases with "dirty" data and evaluating both accuracy and efficiency.To ensure both training efficiency and SQL generation accuracy, this study follows the principles of "high quality, executable, and low redundancy" for data filtering and optimization. For the training of the BIRD and SPIDER benchmarks, we prioritize execution validity checks. Batch execution of reference SQL queries revealed that some samples returned empty results, which, if used for RL training, would fail to provide valid reward signals and could lead to learning biases or "reward hacking."

**Baselines.** We compare our MTIR-SQL framework against two primary categories of baseline methods. For supervised fine-tuning, we evaluate Qwen2.5-Coder-7B-Instruct (Hui et al., 2024) , a state-of-the-art code generation model fine-tuned on Text-to-SQL datasets using standard cross-entropy loss. For reinforcement learning without tool integration, we implement GRPO on the Qwen3-4B model, using execution accuracy as the reward signal to optimize SQL generation through policy gradient methods. Both baselines use identical training procedures and computational budgets as our proposed framework but lack access to intermediate execution feedback during generation, allowing us to isolate the contribution of Multi-turn Tool-Integrated Reasoning.

**Experimental Details.** We conduct experiments using the Qwen3-Instruct model. During training and inference, we adopt database prompts from CodeS (Li et al., 2024b) and SQL-R1 (Ma et al., 2025), which provide curated schema components, values, and metadata, and have demonstrated competitive performance on the BIRD benchmark. We employ algorithms such as PPO and GRPO within the RL-Factory framework (Chai et al., 2025).The training configuration uses a batch size of 64 and a learning rate of 1e-6. During the rollout phase, we sample 5 outputs for each input at temperature T = 0.6, set the maximum sequence length to 8192, and the maximum number of interactions to N = 6. During inference, we apply greedy decoding (T = 0.0). We use SQLite as the SQL executor to obtain execution feedback. The feedback includes column headers and cell values for up to 10 rows. All experiments are conducted on a system equipped with 8 NVIDIA A100 GPUs.

Table 1: Comprehensive comparison of BIRD Dev (EX%) scores. The "OSS" column indicates whether the model is open-source (✓) or proprietary.

| Model | Size | OSS | BIRD Dev (EX%) |
|---|---|---|---|
| ***Models Under 10B Parameters*** | | | |
| *Base Models* | | | |
| DPSK-Coder-6.7B-Instruct Guo et al. (2024) | 6.7B | ✓ | 43.1 |
| Qwen3-4B | 4B | ✓ | 48.1 |
| Qwen2.5-Coder-3B-Instruct Hui et al. (2024) | 3B | ✓ | 50.5 |
| Qwen3-8BMa et al. (2025) | 8B | ✓ | 50.8 |
| Qwen2.5-Coder-7B-Instruct Hui et al. (2024) | 7B | ✓ | 50.9 |
| OpenCoder-8B-Instruct Huang et al. (2025) | 8B | ✓ | 37.5 |
| *SQL-Specific* | | | |
| SFT CodeS-7B Li et al. (2024b) | 7B | ✓ | 57.2 |
| Think2SQL-3B Papicchio et al. (2025) | 3B | ✓ | 50.0 |
| SQL-R1-3B Ma et al. (2025) | 3B | ✓ | 54.6 |
| Think2SQL-7B Papicchio et al. (2025) | 7B | ✓ | 56.1 |
| CogniSQL-R1-Zero-7BGajjar et al. (2025) | 7B | | 59.2 |
| ReEx-SQL-7BMa et al. (2025) | 7B | | 64.9 |
| SQL-R1-7B Ma et al. (2025) | 7B | ✓ | 66.6 |
| Alpha-SQL+ Qwen2.5-Coder-7BGajjar et al. (2025) | 7B | ✓ | 66.8 |
| Arctic-Text2SQL-R1-7BYao et al. (2025) | 7B | ✓ | 68.9 |
| ***Models 10B–100B Parameters*** | | | |
| *Base Models* | | | |
| Granite-20B-Code-Instruct Mishra et al. (2024) | 20B | ✓ | 34.0 |
| Starcoder-15B-Instruct Lozhkov et al. (2024) | 15B | ✓ | 38.5 |
| DPSK-Coder-V2-InstructDeepSeek-AI et al. (2024) | 16B | ✓ | 44.6 |
| Qwen3-14BMa et al. (2025) | 14B | ✓ | 51.8 |
| Codestral-22B team (2024) | 22B | ✓ | 52.7 |
| Qwen2.5-Coder-14B-Instruct Hui et al. (2024) | 14B | ✓ | 61.5 |
| *SQL-Specific* | | | |
| SFT Code5-15B Li et al. (2024b) | 15B | ✓ | 58.5 |
| Reasoning-SQL-14B Pourreza et al. (2025) | 14B | | 64.2 |
| SQL-R1-14B Ma et al. (2025) | 14B | ✓ | 67.1 |
| Arctic-Text2SQL-R1-14BYao et al. (2025) | 14B | ✓ | 70.1 |
| Arctic-Text2SQL-R1-32BYao et al. (2025) | 14B | ✓ | 70.5 |
| ***Large-scale Models (> 100B or Proprietary)*** | | | |
| *Base Models* | | | |
| Mistral Baseline Li et al. (2023) | 123B | ✓ | 53.5 |
| DeepSeek-V3 DeepSeek-AI et al. (2025) | 671B | ✓ | 63.2 |
| *SQL-Specific* | | | |
| SuperSQL (NLSQL-1360) Li et al. (2024a) | – | | 58.5 |
| ChatGPT + CoT Li et al. (2023) | – | | 64.6 |
| MCTS-SQL+GPT-4Li et al. (2023) | – | | 69.4 |
| OpenSearch-SQL+GPT-4oXie et al. (2025) | – | | 69.3 |
| CHASE-SQL+Gemini 1.5 Pourreza et al. (2024) | – | | 73.1 |
| **MTIR-SQL + Qwen3-4B (Ours)** | **4B** | ✓ | **64.4** |
| **MTIR-SQL + Qwen3-8B (Ours)** | **8B** | ✓ | **64.6** |
| **MTIR-SQL + Qwen3-14B (Ours)** | **14B** | ✓ | **68.1** |

## 4.2 MAIN RESULT

**Performance on Main Benchmarks.** In Table 1, we present a comprehensive comparison of MTIR-SQL against state-of-the-art baselines across varying parameter scales. In the compact model regime (under 10B parameters), MTIR-SQL demonstrates exceptional parameter efficiency. Specifically, our **MTIR-SQL (4B)** achieves an execution accuracy of **64.4%** on the BIRD Dev set. Remarkably, despite having significantly fewer parameters, it outperforms robust open-source baselines such as SFT CodeS-7B (57.2%) and Think2SQL-7B (56.1%), and matches the performance of proprietary pipelines like ChatGPT + CoT (64.6%). While recent reinforcement learning-based models like Arctic-Text2SQL-R1-7B achieve higher scores, our model offers a superior trade-off between com-

Table 2: Performance comparison of reasoning paradigms on benchmarks with pass@1.

| Reasoning Paradigm | Training Type | BIRD Dev EX (%) | SPIDER Dev EX (%) | SPIDER Test EX (%) |
|---|---|---|---|---|
| Direct Output | – | 46.9 | 69.2 | 70.8 |
| Standard Reasoning | – | 48.1 | 72.5 | 72.9 |
| Tool-Integrated Reasoning | – | 47.6 | 71.1 | 73.6 |
| Standard Reasoning | GRPO | 58.9 | 78.2 | 79.1 |
| Multi-turn TIR | PPO | 58.2 | 77.2 | 79.2 |
| Multi-turn TIR | GRPO | 60.3 | 80.1 | 81.4 |
| **Multi-turn TIR** | **GRPO-Filter** | **63.1** | **82.4** | **83.4** |

putational cost and performance, effectively bridging the gap between lightweight deployment and high-precision reasoning.

To assess the scalability of our framework, we extended MTIR-SQL to larger scales with 8B and 14B parameters. As shown in the 10B-30B parameter section of Table 1, the performance of our method consistently improves with model size. Most notably, **MTIR-SQL (14B)** achieves **68.1%**, surpassing competitive peers including SQL-R1-14B (67.1%) and Reasoning-SQL (64.2%). This result highlights the effectiveness of our training strategy in eliciting complex SQL generation capabilities, allowing our model to outperform other advanced RL-based methods within the same parameter class.

Finally, we compare MTIR-SQL with large-scale proprietary models. It is worth noting that MTIR-SQL (4B) already surpasses the massive DeepSeek-V3 (63.2%), illustrating that specialized training can yield better domain-specific results than general-purpose giant models. Furthermore, our 14B model approaches the performance of sophisticated multi-agent systems such as OpenSearch-SQL + GPT-4o (69.3%) and CHASE-SQL+Gemini 1.5 (73.1%).

**Performance on Reasoning Paradigms.** In Table 2, we evaluate the performance of different reasoning paradigms on Text-to-SQL benchmarks, specifically focusing on Pass@1 performance across the SPIDER and BIRD datasets. The results highlight the effectiveness of multiturn tool-integrated reasoning. Among the reasoning paradigms, multi-turn TIR with GRPO-Filter leads to the highest performance on both the BIRD and SPIDER benchmarks. The BIRD Dev score of 63.1% represents a significant improvement over standard reasoning and tool-integrated reasoning paradigms, which score 48.1% and 47.6%, respectively. In SPIDER Dev and Test, multi-turn TIR with GRPO filter also excels, achieving 82.4% and 83.4%, respectively, marking a clear advantage over other paradigms.

This reinforces the importance of incorporating execution feedback through multi-turn reasoning for enhancing performance in real-world Text-to-SQL tasks, particularly when dealing with complex databases like SPIDER and BIRD.

Table 3: Robustness Comparison on Spider-DK, Spider-Syn, Spider-Realistic, EHRSQL, and ScienceBenchmark.

| NL2SQL Method | Base Model | Spider-DK | Spider-Syn | Spider-Realistic | EHRSQL | Science Benchmark |
|---|---|---|---|---|---|---|
| *Models Under 7B Parameters* | | | | | | |
| SQL-R1 (Ma et al., 2025) | Qwen2.5-Coder-3B | 70.5 | 66.4 | 71.5 | - | - |
| *Models Under 10B Parameters* | | | | | | |
| OmniSQL (Li et al., 2025b) | Qwen2.5-Coder-7B | 76.1 | 69.7 | 76.2 | 34.9 | 50.2 |
| SQL-R1 (Ma et al., 2025) | Qwen2.5-Coder-7B | 78.1 | 76.7 | 83.3 | - | - |
| Arctic-Text2SQL-R1Yao et al. (2025) | OmniSQL-7B | 81.5 | - | - | 36.7 | 51.8 |
| SQL-o1 (Lyu et al., 2025) | Llama3-8B | 78.7 | 72.6 | 82.7 | - | - |
| *Models Under 30B Parameters* | | | | | | |
| OmniSQL (Li et al., 2025b) | Qwen2.5-Coder-14B | 72.9 | 69.0 | 76.4 | 39.9 | 56.9 |
| SQL-R1 (Ma et al., 2025) | OmniSQL-14B | 79.3 | 78.5 | 86.2 | - | - |
| Arctic-Text2SQL-R1Yao et al. (2025) | OmniSQL-14B | 79.4 | - | - | 40.7 | 58.2 |
| **MTIR-SQL (Ours)** | **Qwen3-4B** | **71.2** | **78.6** | **78.7** | **31.4** | **56.0** |
| **MTIR-SQL (Ours)** | **Qwen3-8B** | **72.9** | **77.2** | **77.4** | **34.4** | **57.0** |
| **MTIR-SQL (Ours)** | **Qwen3-14B** | **76.3** | **81.0** | **81.1** | **36.0** | **60.0** |

**Performance on Cross-domain Benchmark.** We evaluate the robustness and generalization capability of our proposed MTIR-SQL across five challenging benchmarks, spanning perturbation-based datasets (Spider-DK, Spider-Syn, Spider-Realistic) and domain-specific tasks (EHRSQL, ScienceBenchmark), as summarized in Table 3. MTIR-SQL exhibits superior resilience to linguistic variations and domain shifts. Notably, on **Spider-Syn**, our approach achieves state-of-the-art performance, with the Qwen3-14B backbone reaching 81.0%, surpassing the competitive SQL-R1 (OmniSQL-14B) score of 78.5%. Furthermore, in the highly specialized **ScienceBenchmark**, MTIR-SQL establishes a new benchmark high of 60.0%, outperforming Arctic-Text2SQL-R1 (58.2%) and OmniSQL (56.9%). Even at smaller scales, our method demonstrates remarkable data efficiency; for instance, MTIR-SQL (Qwen3-4B) achieves 78.6% on Spider-Syn, significantly outperforming the similarly sized SQL-R1 (Qwen2.5-Coder-3B) by a margin of 12.2 points. These results validate that MTIR-SQL effectively mitigates performance degradation caused by synonym perturbation and cross-domain transfer.

## 4.3 ABLATION STUDY

**Ablation Study of RL Methods.** To assess the effectiveness of MTIR-SQL, we conducted comparisons against PPO, GRPO, and our improved GRPO-Filter using the Qwen3-4B model. As illustrated in Figure 3 and summarized in Table 2, GRPO converges more rapidly than PPO due to the absence of a critic warm-up phase, but it often suffers from reward collapse in later training stages. PPO, in contrast, provides greater stability but at the cost of slower convergence. Crucially, GRPO-Filter addresses these limitations by selectively filtering low-quality rollouts and removing the KL constraint, thereby stabilizing multi-turn training while achieving substantial performance gains. This demonstrates that our modifications are not merely incremental but essential for enabling robust reinforcement learning in execution-aware Text-to-SQL tasks.

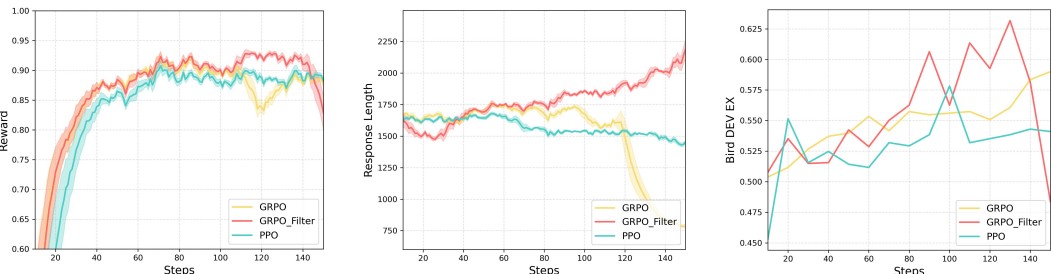

Figure 3: Comparing the impact of different RL Methods on training and performance.

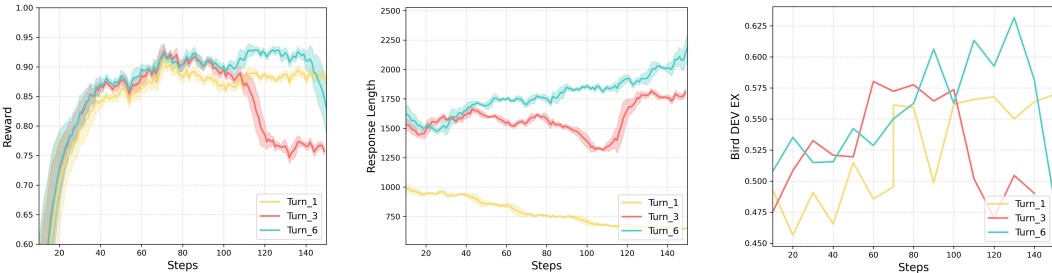

Figure 4: Comparing the impact of different max turns on training and performance.

**Ablation Study on Max Turns.** We further analyzed the impact of maximum tool calling turns by conducting experiments with settings of 1, 3, and 6. The training curves are shown in Figure 4, and the quantitative results are summarized in Table 2. The main findings are as follows: (1) Larger Max Turns generally lead to higher final rewards and stronger overall performance. More turns provide the model with additional opportunities to explore, optimize responses, and execute complex tasks; (2) However, excessive turns (such as 6) may also result in training instability, occasionally causing reward saturation or collapse phenomena; (3) Although Max Turns = 1 demonstrates faster conver-

gence, due to severely limited interaction flexibility, there exists a gap between the final performance and optimal values.

Table 4: Ablation of Reward Components for MTIR-SQL on BIRD Dev.

| Reward Components | BIRD Dev (EX %) |
|---|---|
| MTIR-SQL | 63.1 |
| w/o $R_{\text{format}}$ | 62.3 ↓ (0.8) |
| w/o $R_{\text{exec}}$ | 59.4 ↓ (3.9) |
| w/o $R_{\text{result}}$ | 58.8 ↓ (4.3) |

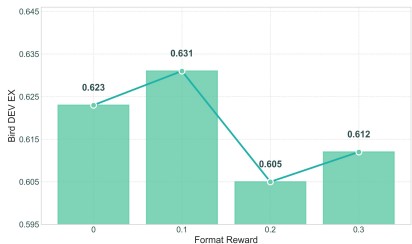

Figure 5: Ablation of Reward Components for MTIR-SQL on BIRD Dev Reward.

**Ablation Study on Reward Design.** This ablation study evaluates the impact of removing each reward component on the model's performance using the BIRD Development set:

- $R_{\mathbf{f}}$ **(Format Reward):** We analyze the sensitivity of the model to the format reward weight, as visualized in Figure 5. Setting the format reward coefficient to a moderate value (0.1) improves performance from 62.3% to 63.1%, representing a 0.8% increase compared to the baseline (0). However, further increasing the weight to 0.2 leads to a significant performance drop to 60.5% (a 4.1% decline), indicating that excessive format rewards can negatively impact the model by over-constraining generation. A slight recovery is observed at 0.3 (61.2%), reinforcing the conclusion that a balanced format reward is beneficial but should not be overemphasized.

- $R_{\mathbf{e}}$ **(Execution Reward):** Removing the execution reward results in the largest performance drop, from 63.1% to 59.4% (a 3.9% decrease), highlighting its crucial role in the natural language-to-SQL conversion process. Without execution-based feedback, the model struggles to make accurate predictions.

- $R_{\mathbf{r}}$ **(Result Reward):** Excluding the result reward leads to a smaller decline in performance, from 63.1% to 58.8% (a 4.3% drop), underlining its importance in ensuring the functional correctness of the model's SQL queries.

In conclusion, removing any reward—particularly $R_{\mathbf{f}}$—significantly hampers the model's performance. This underscores the necessity of a balanced reward system that integrates execution feedback, exploration, and result accuracy for optimal performance.

## 5 CONCLUSION

We propose **MTIR-SQL**, a novel reinforcement learning framework for complex multi-turn SQL generation tasks. MTIR-SQL's central innovation resides in its feedback-driven reasoning approach, where execution results inform subsequent reasoning iterations, creating a self-correcting mechanism that substantially improves generation stability and query accuracy. We extend GRPO with **trajectory filtering** to mitigate distribution drift and remove KL divergence constraints to enhance learning efficiency. Experimental results demonstrate MTIR-SQL's effectiveness: achieving **64.4%** accuracy on BIRD-SQL and **84.6%** execution accuracy on SPIDER with a 4B-parameter model, significantly outperforming baseline methods and advancing state-of-the-art in Text-to-SQL generation.

# 6 ETHICS STATEMENT

This study uses publicly available datasets (BIRD and SPIDER) and does not involve private or confidential data. No human participants are included, and we ensure fairness and transparency in our model's design and deployment.

# 7 REPRODUCIBILITY STATEMENT

The model code, datasets, and experimental setup are available upon request. Detailed instructions for reproducing our experiments are provided to ensure transparency and facilitate further research in the Text-to-SQL domain.

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

## A USE OF LLMs

In this work, we use Large Language Models (LLMs) for text refinement and grammar checking. LLMs help improve the clarity, coherence, and grammatical accuracy of the generated content, ensuring the final text meets academic standards. Their use is limited to enhancing written content, without influencing the research methodology or decision-making process.

## B LLM RESPONSE ROLLOUT WITH MULTI-TURN SQL EXCURSIONS CALLS

The algorithm describes the response generation process of a generative model (e.g., LLM) based on multi-turn interactions. The core idea of the algorithm is to progressively generate a response sequence based on the user's input and previous responses. In each generation step, the model evaluates the current output and interacts with external tools for validation (e.g., executing SQL queries). The results returned by the tool are then integrated into the generated response. The entire process is conducted within a maximum action budget to ensure that the final output meets the problem's requirements and is validated for accuracy. After each round, the model adjusts its output based on the results, continuing until a complete response is achieved or the budget limit is reached.

---

**Algorithm 1** LLM Response Rollout with Multi-Turn SQL Execution Tool Calls

---

**Require:** Input query $x$, policy model $\pi_\theta$, SQL execution tool $\mathcal{T}$, maximum action budget $B$.
**Ensure:** Final response $y$.

1: Initialize rollout sequence $y \leftarrow \emptyset$
2: Initialize action count $b \leftarrow 0$
3: **while** $b < B$ **do**
4:     Initialize current action LLM rollout sequence $y_b \leftarrow \emptyset$
5:     **while** True **do**
6:         Generate response token $y_t \sim \pi_\theta(\cdot|x, y, y_b)$
7:         Append $y_t$ to rollout sequence $y_b \leftarrow y_b + y_t$
8:         **if** $y_t \in \langle\text{tool call}\rangle, \langle\text{tool response}\rangle, \langle\text{eos}\rangle$ **then**
9:             **break**
10:         **end if**
11:     **end while**
12:     $y \leftarrow y + y_b$
13:     **if** $\langle\text{tool call}\rangle$ detected in $y_b$ **then**
14:         Extract SQL query $q \leftarrow \text{Parse}(y_b, \langle\text{tool call}\rangle, \langle\text{tool call}\rangle)$
15:         Retrieve SQL query results $d \leftarrow \mathcal{T}(q)$
16:         Insert $d$ into rollout $y \leftarrow y + \langle\text{tool response}\rangle d \langle\text{tool response}\rangle$
17:     **else if** $\langle\text{tool response}\rangle$ detected in $y_b$ **then**
18:         **return** final generated response $y$
19:     **else**
20:         Ask for rethink $y \leftarrow y + \langle\text{My action is not correct. Let me rethink.}\rangle$
21:     **end if**
22:     Increment action count $b \leftarrow b + 1$
23: **end while**
24: **return** final generated response $y$

---

# C ADDITIONAL RESULTS AND ANALYSIS

## C.1 ANALYSIS AND COMPARISON OF BASE LLMs

As presented in Table 5, we conducted a comprehensive evaluation of different base LLMs to investigate the effectiveness of our proposed training framework. We selected strong code-specialized models (Qwen2.5-Coder series) and NL2SQL-specific models (OmniSQL series) as baselines, comparing them against the Qwen3 (Thinking Mode) series.

A critical observation from the baseline results is the **initial performance disparity**. As shown in the middle section of Table 5, the vanilla Qwen3 models in Thinking Mode exhibited relatively modest performance, significantly lagging behind the Qwen2.5-Coder and OmniSQL counterparts. For instance, on the challenging BIRD (Dev) benchmark, the Qwen3-4B base model achieved an accuracy of only 48.1%, which is notably lower than the 50.5% of Qwen2.5-Coder-3B and substantially behind the domain-specific OmniSQL-7B (61.5%). This suggests that the inherent NL2SQL capability of the Qwen3 (Thinking Mode) backbone is initially weak and suboptimal for direct deployment in complex reasoning scenarios.

However, the integration of our **MTIR-SQL** framework yields a transformative improvement. Despite the weak initialization, the models fine-tuned with MTIR-SQL demonstrated remarkable performance gains across all metrics. Specifically:

- On the Spider (Dev) set, MTIR-SQL enabled the Qwen3-4B model to surge from 72.3% to **82.4%**, a substantial absolute improvement of **10.1%**.

- Similarly, on the BIRD (Dev) benchmark, the 14B variant improved from a baseline of 51.8% to **67.2%**, achieving a massive gain of **15.4%**.

These results highlight a pivotal insight: while the base Qwen3 models do not possess state-of-the-art capabilities out-of-the-box, they exhibit exceptional *plasticity* and potential when guided by our proposed method. The significant delta between the base and fine-tuned results confirms that MTIR-SQL successfully activates the model's latent reasoning abilities, allowing a weaker backbone to achieve competitive performance comparable to, or even exceeding, larger models trained with standard supervised fine-tuning. This underscores the efficacy of our training strategy in bridging the gap between weak initialization and high-performance execution.

Table 5: Comparison of different base LLMs on Spider and BIRD benchmark with Greedy Search. Note that "Thinking Mode" refers to the vanilla Qwen3 behavior without our specific training.

| Base Model | Spider (Dev) | Spider (Test) | BIRD (Dev) |
|---|---|---|---|
| Qwen2.5-Coder-3B | 77.0 | 77.2 | 50.5 |
| Qwen2.5-Coder-7B | 73.4 | 82.2 | 50.9 |
| Qwen2.5-Coder-14B | 78.1 | 86.6 | 61.5 |
| Qwen2.5-Coder-32B | 77.7 | 87.5 | 64.5 |
| OmniSQL-7B | 81.2 | 87.9 | 61.5 |
| OmniSQL-14B | 81.4 | 88.3 | 64.2 |
| OmniSQL-32B | 80.9 | 87.6 | 64.5 |
| Qwen3-4B (Thinking Mode) | 72.3 | 72.8 | 48.1 |
| Qwen3-8B (Thinking Mode) | 73.5 | 76.1 | 50.8 |
| Qwen3-14B (Thinking Mode) | 75.9 | 76.2 | 51.8 |
| **MTIR-SQL + Qwen3-4B** | **82.4** | **83.4** | **63.1** |
| **MTIR-SQL + Qwen3-8B** | **83.6** | **84.2** | **63.6** |
| **MTIR-SQL + Qwen3-14B** | **86.7** | **87.2** | **67.2** |

## C.2 ANALYSIS AND COMPARISON OF EFFICIENCY

To evaluate the practical deployability of our proposed framework, we conduct a comprehensive analysis of inference efficiency on the BIRD-dev dataset. We focus on three key metrics: *Latency per Question*, *Total Tokens per Question* (including input prompt and output completion), and *Execution Accuracy*. Additionally, we investigate the *Tool Call Frequency* to understand the reasoning behavior of our model. The results are summarized in Table 6.

**Trade-off between Accuracy and Overhead.** As illustrated in Table 6, existing methods often struggle to balance performance with computational cost. While **CHESS** achieves a respectable accuracy of 61.5%, it incurs a prohibitive computational penalty, requiring an average of 251.3 seconds and over 320K tokens per query. This suggests that its multi-turn reasoning or agent-based retrieval mechanisms, though effective, are inefficient for real-time applications. Conversely, **Qwen2.5-Coder-7B** offers the lowest latency (0.3s) but lags significantly in semantic correctness (58.2%), indicating a limitation in handling complex schema linking without sufficient reasoning depth.

**Superiority of MTIR-SQL.** Our proposed **MTIR-SQL** series demonstrates a superior efficiency-performance frontier. Notably, **MTIR-SQL-14B** establishes an accuracy of **67.2%**, outperforming the strong baseline SQL-R1-7B by a margin of 3.5%. More importantly, this performance gain does not come at the cost of efficiency. MTIR-SQL-14B consumes only **1.7K tokens** on average—the lowest among all compared methods—while maintaining a latency of 0.5s. This counter-intuitive result, where a larger model uses fewer tokens, indicates that the MTIR strategy effectively aligns the model to generate concise, precise SQL queries directly, reducing the need for verbose chain-of-thought reasoning or iterative self-correction.

**Strategic Tool Utilization.** A distinctive feature of our framework is the capability for autonomous tool interaction. As shown in the "Tool Call / Question" column of Table 6, the MTIR-SQL models exhibit a highly efficient usage pattern, averaging between **1.31 and 1.42 tool calls** per question. This low frequency is particularly revealing: it suggests that the Reinforcement Learning refinement has taught the model to invoke external tools (e.g., for schema state verification or preliminary execution) *selectively* rather than indiscriminately. Unlike redundant agent loops that inflate latency, our model executes tools only when essential for resolving ambiguity. This "precision-first" behavior explains how MTIR-SQL achieves high accuracy (67.2%) with minimal latency overhead (0.5s), effectively validating that intelligent tool use can enhance performance without compromising deployment efficiency.

Furthermore, even our smaller variants, **MTIR-SQL-4B** and **8B**, exhibit competitive accuracy (63.1% and 63.6%) with minimal latency overhead, proving that our training methodology is model-agnostic and highly scalable. Overall, MTIR-SQL provides the most viable solution for production environments where both high precision and low latency are critical.

Table 6: Efficiency comparison of different NL2SQL methods on BIRD-dev dataset.

| NL2SQL Method | Candidate Selection | Latency (s) / Question | Total Tokens (K) Question | Tool Call Question | Accuracy (%) |
|---|---|---|---|---|---|
| Qwen2.5-Coder-7B | Greedy Search | 0.3 | 2.5 | - | 58.2 |
| XiYan-SQL-7B | Greedy Search | 0.5 | 4.1 | - | 62.1 |
| CHESS | Greedy Search | 251.3 | 320.8 | - | 61.5 |
| SQL-R1-7B | Greedy Search | 0.4 | 3.1 | - | 63.7 |
| **MTIR-SQL-4B (Ours)** | **Greedy Search** | **0.5** | **2.9** | **1.34** | **63.1** |
| **MTIR-SQL-8B (Ours)** | **Greedy Search** | **0.4** | **2.0** | **1.31** | **63.6** |
| **MTIR-SQL-14B (Ours)** | **Greedy Search** | **0.5** | **1.7** | **1.42** | **67.2** |

## D PROMPT AND CASE STUDY

Prompts used while training and several cases are presented as follows.

### D.1 SYSTEM PROMPT

**System Prompt of Tool**

**##Tools**

You may call one or more functions to assist with the user query.

You are provided with function signatures within <tools></tools> XML tags:

```
<tools>
  {"name": "sql-execute_sql_query", "description": "
     Execute SQL query and return partial results
     containing column names (maximum 10 records).

   Args:db_name (str): The name of the database.
     sql (str): The SQL query to execute.

   Returns:Dict[str, Union[List[Dict], Dict, None]]: A
       dictionary containing 'columns' and 'data' of the
       query (maximum of 10 records).

   Raises: TimeoutError: If the query execution exceeds
       the timeout.
     sqlite3.Error: If an error occurs during the query
         execution.
   ",
     "parameters": {
       "type": "object",
       "properties": {
         "db_name": {"title": "Db Name", "type": "string"},
         "sql": {"title": "Sql", "type": "string"}
       },
       "required": ["db_name", "sql"]
     }
  }
</tools>
```

For each function call, return a JSON object with function name and arguments within <tool_call></tool_call> XML tags:

```
<tool_call>
  {"name": <function-name>, "arguments": <args-json-object
     >}
</tool_call}
```

## D.2 USER PROMPT

---

### Prompts used while training and evaluating

You are a helpful SQL expert assistant. You should first think about how to write the SQL query by analyzing the question, database schema, and external knowledge, then validate your SQL with the tool until it is correct. Finally, you provide the final SQL query in `<answer> </answer>`.

**Task Configuration**
**Database Engine:** SQLite
**Database:** {db_id}
**Database Schema:** {schema}
**User Question:** {external_knowledge} + {question}?
**Requirements**

1. **Precision:** Make sure you only output the information that is asked in the question. If the question asks for a specific column, make sure to only include that column in the SELECT clause, nothing more.

2. **Completeness:** The generated query should return all of the information asked in the question without any missing or extra information.

3. **Correctness:** Before generating the final SQL query, please think through the steps of how to write the query. Validate your SQL through tool testing.

**Output Format:**
**Important:** Use EITHER thinking + tool calls OR thinking + final answer. Do not mix the structures.
**Option A (when validation needed):**

```
<think> Your analysis... </think>
[Tool calls for validation]
```

**Option B (final answer):**

```
<think> Your final analysis... </think>
<answer>
```sql
YOUR_SQL_QUERY
</answer>
```

---

## D.3 CASE STUDY

---

**Case1 Single Turn**

USER QUESTION

Exclusively virtual refers to Virtual = 'F'. How many schools with an average score in Math greater than 400 in the SAT test are exclusively virtual?

MODEL RESPONSE

<think>
Okay, let's tackle this problem. The user is asking for the number of schools that are exclusively virtual (Virtual = 'F') and have an average math score greater than 400 in the SAT test.... So the query should be correct.
</think>
<tool_call>

```
{
  "name": "sql-execute_sql_query",
  "arguments": {
    "db_name": "california_schools",
    "sql": "SELECT COUNT(*) FROM satscores JOIN schools ON satscores
        .cds = schools.CDSCode WHERE schools.Virtual = 'F' AND
        satscores.AvgScrMath > 400;"
  }
}
```

</tool_call>
<tool_response>

```
The result is:{
  "columns": [
    "COUNT(*)"
  ],
  "data": [
    {
      "COUNT(*)": 4
    }
  ]
}
```

</tool_response>
<think>
Okay, let me check the user's question again. They want the number of schools that are exclusively virtual (Virtual = 'F') and have an average math score greater than 400 in the SAT.....The result from the tool is 4, so that's the answer.
</think>
<answer>
```sql

**SELECT COUNT**(*) **FROM** satscores **JOIN** schools **ON** satscores.cds =
    schools.CDSCode **WHERE** schools.Virtual = 'F' **AND** satscores.
    AvgScrMath > 400;

```

</answer>

---

**Case2 Multi-Turn**

USER QUESTION

calcium refers to element = 'ca'; label = '+' mean molecules are carcinogenic; label = '-' means molecules are non-carcinogenic; MAX(label).Among the molecules with element Calcium, are they mostly carcinogenic or non carcinogenic?

MODEL RESPONSE

**Turn 1**

<think>
Alright, let's solve this problem. The user wants to know whether molecules containing calcium element (element 'Ca') are mostly carcinogenic or not. Label '+' means carcinogenic, '-' means non-carcinogenic. So, we need to find the maximum label value for molecules containing calcium... So the final SQL query will be to select the maximum label from molecules that contain calcium atoms.
</think>
<tool_call>

```
{
    "name": "sql-execute_sql_query",
    "arguments": {
      "db_name": "toxicology",
      "sql": "SELECT MAX(m.label) AS max_label FROM molecule m JOIN
          atom a ON m.molecule_id = a.molecule_id WHERE a.element = '
          Ca';"
    }
}
```

</tool_call>
<tool_response>

```
The result is:: {
    "columns": [
      "max_label"
    ],
    "data": [
      {
        "max_label": null
      }
    ]
}
```

</tool_response>
**Turn 2**
<think>
Hmm, the SQL query result returns max_label as null. This is unexpected. Let me check the query again. Maybe the label in the molecule table is stored as a string, but perhaps the query didn't correctly select the label... Let me try the corrected query.
</think>
<tool_call>

```
{
    "name": "sql-execute_sql_query",
    "arguments": {
      "db_name": "toxicology",
      "sql": "SELECT MAX(m.label) AS max_label FROM molecule m JOIN
          atom a ON m.molecule_id = a.molecule_id WHERE a.element = '
          CA';"
    }
}
```

</tool_call>
<tool_response>

```
The result is: {
  "columns": [
    "max_label"
  ],
  "data": [
    {
      "max_label": null
    }
  ]
}
```

</tool_response>
**Turn 3**
<think>
Wait, the result is still null. Let me think again. Maybe the label is stored as a string in the molecule table, but the query didn't select it correctly. Let me try the corrected query.
</think>
<tool_call>

```
{
  "name": "sql-execute_sql_query",
  "arguments": {
    "db_name": "toxicology",
    "sql": "SELECT MAX(m.label) AS max_label FROM molecule m JOIN
        atom a ON m.molecule_id = a.molecule_id WHERE a.element = '
        ca';"
  }
}
```

</tool_call>
<tool_response>

```
The result is: {
  "columns": [
    "max_label"
  ],
  "data": [
    {
      "max_label": "-"
    }
  ]
}
```

</tool_response>
**Turn 4**
<think>
Okay, so after adjusting the query to use 'ca' as the element, the result is that the max_label is '-'. That means among the molecules with calcium atoms, the maximum label is '-', which indicates they are non-carcinogenic. Therefore, the answer should be that they are mostly non-carcinogenic.
</think>
<answer>
"'sql

```
SELECT MAX(m.label) AS max_label FROM molecule m JOIN atom a ON m.
    molecule_id = a.molecule_id WHERE a.element = 'ca';
```

"'

</answer>

## Case3 Multi-Turn

USER QUESTION

What is the number of SAT test takers of the schools with the highest FRPM count for K-12 students?

MODEL RESPONSE

**Turn 1**
<think>
Alright, let me solve this problem. The user is asking about the number of SAT test takers at schools with the highest FRPM count for K-12 students. First, I need to determine which tables are involved.... Let me verify using the tool.
</think>
<tool_call>

```
{
  "name": "sql-execute_sql_query",
  "arguments": {
    "db_name": "california_schools",
    "sql": "SELECT SUM(s.NumTstTakr) FROM satscores s JOIN frpm f ON
        s.cds = f.CDSCode WHERE f.\"FRPM Count (K-12)\" = (SELECT
        MAX(\"FRPM Count (K-12)\") FROM frpm);"
  }
}
```

</tool_call>
<tool_response>

```
{
  "columns": [
    "SUM(s.NumTstTakr)"
  ],
  "data": [
    {
      "SUM(s.NumTstTakr)": null
    }
  ]
}
```

</tool_response>
**Turn 2**
<think>
Hmm, the SQL query returned a sum result of null. This is not right. Let me check the query again. Maybe there's an error in the column names or table names... Let me test again.
</think>
<tool_call>

```
{
  "name": "sql-execute_sql_query",
  "arguments": {
    "db_name": "california_schools",
    "sql": "SELECT SUM(s.NumTstTakr) FROM satscores s JOIN frpm f ON
        s.cds = f.CDSCode WHERE f.\"FRPM Count (K-12)\" = (SELECT
        MAX(\"FRPM Count (K-12)\") FROM frpm);"
  }
}
```

</tool_call>
<tool_response>

```
{
  "columns": [
    "SUM(s.NumTstTakr)"
```

```
        ],
        "data": [
            {
                "SUM(s.NumTstTakr)": null
            }
        ]
    }
```

</tool_response>
**Turn 3**
<think>
Hmm, the result is still null. This is strange. Maybe there's an issue with the query structure.
I need to check if the column name in the frpm table is really named "FRPM Count (K-12)"
or if there's a spelling error. Let me try again.
</think>
<tool_call>

```
{
    "name": "sql-execute_sql_query",
    "arguments": {
        "db_name": "california_schools",
        "sql": "SELECT SUM(s.NumTstTakr) FROM satscores s JOIN frpm f ON
            s.cds = f.CDSCode WHERE f.\"FRPM Count (K-12)\" = (SELECT
            MAX(\"FRPM Count (K-12)\") FROM frpm);"
    }
}
```

</tool_call>
<tool_response>

```
{
    "columns": [
        "SUM(s.NumTstTakr)"
    ],
    "data": [
        {
            "SUM(s.NumTstTakr)": null
        }
    ]
}
```

</tool_response>
**Turn 4**
<think>
Maybe I should first check what the maximum value of FRPM Count (K-12) in the frpm
table is. Let me write a query to get that value.
</think>
<tool_call>

```
{
    "name": "sql-execute_sql_query",
    "arguments": {
        "db_name": "california_schools",
        "sql": "SELECT SUM(s.NumTstTakr) FROM satscores s JOIN frpm f ON
            s.cds = f.CDSCode WHERE f.\"FRPM Count (K-12)\" = (SELECT
            MAX(\"FRPM Count (K-12)\") FROM frpm);"
    }
}
```

</tool_call>
<tool_response>

```
{
    "columns": [
        "SUM(s.NumTstTakr)"
```

```
      ],
      "data": [
        {
          "SUM(s.NumTstTakr)": null
        }
      ]
    }
```

</tool_response>
<think>
Maybe there's an issue with the column names or the tables not joining correctly. Let me think again. Maybe I should check if the satscores table has data for schools in the frpm table.... I think this should work. Let me use this as the final query.
</think>
<tool_call>

```
    {
      "name": "sql-execute_sql_query",
      "arguments": {
        "db_name": "california_schools",
        "sql": "SELECT SUM(s.NumTstTakr) FROM satscores s JOIN (SELECT
            CDSCode FROM frpm WHERE \"FRPM Count (K-12)\" = (SELECT MAX
            (\"FRPM Count (K-12)\") FROM frpm)) AS top_frpm ON s.cds =
            top_frpm.CDSCode;"
      }
    }
```

</tool_call>
<tool_response>

```
    {
      "columns": [
        "SUM(s.NumTstTakr)"
      ],
      "data": [
        {
          "SUM(s.NumTstTakr)": 217547
        }
      ]
    }
```

</tool_response>
**Turn 5**
<think>
The tool response shows a result of 217547, which seems to be a valid number.
</think>
<answer>
‴'sql

```sql
SELECT SUM(s.NumTstTakr) FROM satscores s JOIN (SELECT CDSCode FROM
    frpm WHERE "FRPM_Count_(K-12)" = (SELECT MAX("FRPM_Count_(K-12)"
    ) FROM frpm)) AS top_frpm ON s.cds = top_frpm.CDSCode;
```

‴'
</answer>

