# OpenReview forum: "MTIR-SQL:Multi-turn Tool-Integrated  Reasoning Reinforcement Learning for Text-to-SQL"
_ICLR.cc/2026/Conference — Submitted to ICLR 2026_

### Official Review · Reviewer_EgQd · 2025-10-25

**Soundness:** 2
**Presentation:** 3
**Contribution:** 1
**Rating:** 4
**Confidence:** 4

**Summary:**

This paper introduces MTIR-SQL, a multi-turn RL NL2SQL framework with SQL execution feedback. It is trained with a modified GRPO algorithm with trajectory filtering and without KL regularization. The experiment results on the Qwen3-4B model suggest good accuracy on BIRD and SPIDER benchmarks.

**Strengths:**

- The empirical results on BIRD dev seem to be strong for a 4B model.
- The paper addresses a well-motivated NL2SQL problem.

**Weaknesses:**

- **Limited Methodological Novelty:** The paper's primary weakness is its limited methodological contribution. The core ideas are largely applications or incremental modifications of existing techniques.
    - The MTIR framework closely resembles the self-correction/multi-turn reasoning workflows already established in prior work (e.g., DART-SQL, CHASE-SQL, Arctic-Text2SQL-R1). Integrating this loop with RL, while effective, appears to be a straightforward incremental step.
    - The extensions to the GRPO algorithm, such as removing KL regularization and adding trajectory filtering, are common practices in modern RL training pipelines (e.g., verl, DAPO, rStar-Math). Any potential novelty in the *design* of the filtering functions is unfortunately not discussed, making the contribution in this area unclear.
- **Insufficient Baseline Comparisons:** The performance evaluation lacks a fair comparison against key baselines. While Tables 1-3 show the 4B model outperforming some larger models, they omit direct comparisons against current state-of-the-art (SOTA) RL-based models from the BIRD benchmark leaderboard, such as arctic-r1. This omission makes it difficult to properly situate the paper's performance claims.
- **Lack of Scalability Analysis:** The empirical validation is limited to a single 4B model. The paper provides no analysis of how the methodology performs with models of different scales (e.g., 7B, 14B, or 32B), making it difficult to judge the generalizability of the approach.
- **Contribution Feels More Engineering than Research:** The work reads more like a strong engineering report, demonstrating a successful combination of existing components, than a fundamental research paper. As a result, I am concerned that it offers limited new insights or generalizable lessons for the ICLR community.

**Questions:**

- What is “retrieval-based token masking” in line 99, page 2? There is no explanation for this term in the rest of this paper.
- What is the filtering function for selective rollout filtering?
- In figure 3, given the fact that GRPO_Filter starts to collapse after step 140, is it possible that GRPO finally outperform GRPO_Filter? What makes GRPO_Filter collapse earlier than the original GRPO method?

---

> ### Author Response · Authors · 2025-11-26
> **Response to Reviewer EgQd (part1)**
>
> We sincerely thank the reviewer for the thoughtful review and constructive feedback. We are pleased that the reviewer recognizes the **strong empirical results** of our 4B model on the BIRD benchmark  and that the paper addresses a **well-motivated NL2SQL problem**.
>
> We have carefully addressed all concerns by providing deeper analysis, clarifying the methodological innovations, and extending our empirical results, which we believe significantly strengthens the paper.
>
>
> ### **W1. Limited Methodological Novelty**
>
> We acknowledge that the core components, such as Multi-Turn Tool-Integrated Reasoning (MTIR) and modified GRPO, build upon existing ideas. However, our **primary contribution** is not in introducing isolated novel parts, but in the **systematic integration** of these elements to create a **stable and effective end-to-end RL framework for Text-to-SQL**, a domain where robust multi-turn execution feedback learning remains highly challenging.
>
> * **MTIR-SQL Framework:** We are one of the first to apply the Multi-Turn Tool-Integrated Reasoning paradigm with explicit RL optimization to the Text-to-SQL domain, which is crucial for handling complex, real-world databases like BIRD. Our framework introduces an **execution-aware multi-turn tool-integrated reasoning paradigm** that seamlessly incorporates database execution feedback at **each reasoning step** , enabling dynamic self-correction and progressive refinement , a capability often lacking in prior RL methods.
> * **GRPO Extensions for Stability:** The extensions to GRPO—specifically **removing the KL constraint**  and introducing a **multi-dimensional quality-aware filtering mechanism** —are essential for stability in this complex multi-turn environment. While these concepts exist in general RL literature (e.g., DAPO, SimpleTIR), their specific application and tailored design for Text-to-SQL's unique challenges (e.g., invalid tool call formats, fatal execution errors , and low-variance rollouts) constitute a novel and necessary engineering innovation for this domain. The filtering function, $\mathcal{F}(x,y,G_{x})=\mathbb{I}_{valid}(y)\cdot\mathbb{I}_{div}(G_{x})$, is explicitly designed to address multi-turn interaction patterns and mode collapse.
>
> Furthermore, our framework, built on RL-Factory, opens a **new direction for Agentic RL in Text-to-SQL**. The successful RL-based optimization of multi-turn tool use, as demonstrated by MTIR-SQL, makes it possible to integrate more sophisticated Agent components—such as memory and richer tools (e.g., database decomposition, filtering, and syntax checking)—with end-to-end RL, potentially achieving greater gains with smaller models.
>
> ### **W2. Insufficient Baseline Comparisons**
>
> We appreciate this crucial point. We have revised Section 4.2 and Table 1 to provide a more comprehensive comparison with state-of-the-art RL-based methods in the BIRD benchmark, including those recently published.
>
> | Model | Size | BIRD Dev (EX%) |
> |:---:|:---:|:---:|
> | SQL-R1-7B | 7B | 66.6  |
> | Arctic-Text2SQL-R1-7B | 7B | 68.9  |
> | **MTIR-SQL + Qwen3-4B (Ours)** | **4B** | **64.4**  |
> | **MTIR-SQL + Qwen3-8B (Ours)** | **8B** | **64.6**  |
> | **MTIR-SQL + Qwen3-14B (Ours)** | **14B** | **68.1** |
>
> As shown, our 4B model achieves **64.4%** execution accuracy, which is highly competitive and surpasses several 7B-parameter models. Although Arctic-Text2SQL-R1-7B achieves a higher score, it is important to note the **significant difference in initial model performance**:
>
> | Base Model | BIRD (Dev) (Thinking Mode) |
> |:---:|:---:|
> | OmniSQL-7B | 61.5 |
> | Qwen3-4B | 48.1 |
>
> The Qwen3 base models we used exhibit a **significantly weaker initial NL2SQL capability** compared to domain-specific models like OmniSQL and the base models used by others. The remarkable performance increase of $15.4\%$ for the 14B variant (from $51.8\%$ to $67.2\%$) clearly demonstrates that the **MTIR-SQL framework is exceptionally effective** at activating latent reasoning abilities and rapidly closing the performance gap, making it an **efficient training methodology for smaller, weaker-initialized models**.

---

> ### Author Response · Authors · 2025-11-26
> **Response to Reviewer EgQd (part2)**
>
> ### **W3. Lack of Scalability Analysis**
>
> We agree on the importance of scalability. We have extended our empirical validation to include the results for the 8B and 14B parameter models, as summarized in the revised Table 5 (and Table 1 in the paper).
>
> | Base Model | Spider (Dev) | Spider (Test) | BIRD (Dev) |
> |:---:|:---:|:---:|:---:|
> | **MTIR-SQL + Qwen3-4B** | 82.4  | 83.4  | 63.1 |
> | **MTIR-SQL + Qwen3-8B** | 83.6  | 84.2  | 63.6  |
> | **MTIR-SQL + Qwen3-14B** | 86.7  | 87.2  | 67.2  |
>
> The results show that the performance of MTIR-SQL **consistently improves with model scale**, demonstrating the **generality of our approach**. The MTIR-SQL (14B) model achieves 67.2% execution accuracy on BIRD Dev , **outperforming most other 14B models** (e.g., SQL-R1-14B at 67.1% ) and approaching the performance of much larger, proprietary multi-agent systems.
>
> ### **W4. Contribution Feels More Engineering than Research**
>
> We appreciate the classification of our work as a "strong engineering report." We contend that, in the field of LLMs, **significant advances often stem from robust engineering practices** that successfully integrate theoretical concepts into a stable, high-performance system. Our work, MTIR-SQL, is an example of this. By providing a **stable and standardized framework** for Agentic Reinforcement Learning with tools (built on RL-Factory ), we are making a fundamental contribution that enables future research:
>
> * **Enabling New Research:** Our work demonstrates that complex, multi-turn tool-integrated reasoning workflows—previously tackled with heuristic or supervised methods—can be effectively and robustly optimized with RL, a path not previously achieved in Text-to-SQL.
> * **Generalizable Lessons:** The GRPO-Filter extensions for mitigating distribution drift and collapse in multi-turn tool-use scenarios , a persistent problem in tool-augmented RL, offer a generalizable recipe for building stable Agentic LLMs for other domains beyond SQL.
>
>
> ### **Q1. What is “retrieval-based token masking” in line 99, page 2?**
>
> We sincerely apologize for the confusing and misleading phrasing. This was a terminology error. The correct and more precise term is **"tool response token masking"**. This technique is implemented to stabilize training by masking the tokens corresponding to the execution feedback (tool response) in the loss calculation, preventing the model from over-relying on the tool output during policy updates. This detail has been clarified in the revised paper.

---

> ### Author Response · Authors · 2025-11-26
> **Response to Reviewer EgQd (part3)**
>
> ### **Q2. What is the filtering function for selective rollout filtering?**
>
> The selective rollout filtering mechanism is a **multi-dimensional quality-aware filtering mechanism**  defined by the composite indicator function $\\mathcal{F}(x,y,G_{x}) = \\mathbb{I}\_{valid}(y) \\cdot \\mathbb{I}\_{div}(G_{x})$. We apply two primary criteria:
>
> 1.  **Execution Validity ($\mathbb{I}_{valid})$:** Filters out trajectories with **invalid interaction patterns** , such as sequences where the model performs more than two tool invocations without producing a final answer, or encounters fatal execution errors. This is crucial for efficient learning from multi-turn interactions.
> 2.  **Representation Diversity ($\mathbb{I}_{div})$:** Discards groups of rollouts with **insufficient variance** (i.e., $\sigma_{R}(G_{x}) \leq \epsilon_{\sigma}$) , preventing "mode collapse" and ensuring that the advantage estimation $A(x,y)$ used in the GRPO loss function remains meaningful.
>
> These criteria are elaborated in Section 3.1 of the paper, specifically equations (3) and (4) and the subsequent text.
>
> ### **Q3. Analysis of Figure 3 and GRPO Collapse**
>
> We are extremely grateful for your meticulous observation of Figure 3, which highlights a subtle but critical challenge in modern RL training.
>
> The phenomenon of the GRPO variants collapsing prematurely stems from **Unconstrained Optimization**:
> * The **KL loss constraint** ($KL(\pi_{\theta}||\pi_{ref})$) in vanilla GRPO and PPO is typically used to prevent the trained policy ($\pi_{\theta}$) from deviating too far from the initial reference policy ($\pi_{ref}$).
> * We **removed this KL constraint**  to allow for "more flexible policy updates"  and a potentially higher performance ceiling.
> * The **GRPO's relative optimization property**, coupled with the removal of the KL constraint, makes the training inherently more prone to large, sudden policy updates and distributional shift, leading to a faster and more pronounced collapse than the constrained PPO method.
>
> We address your specific questions as follows:
>
> 1.  **Is it possible that GRPO finally outperforms GRPO\_Filter?**
>     * No. We use the **best performing model checkpoint** before collapse for final evaluation. The comparison shown in Table 2 (63.1% for GRPO-Filter vs. 60.3% for GRPO)  clearly indicates that **GRPO-Filter achieves a significantly higher performance ceiling** before instability sets in. We compare the best performing checkpoints to demonstrate the maximum potential unlocked by each algorithm, not their transient behavior at a fixed step. PPO avoids the early collapse but suffers from slower convergence.
> 2.  **What makes GRPO\_Filter collapse earlier than the original GRPO method?**
>     * In our practical implementation, the **removal of the KL regularization is the primary cause** of the increased instability and earlier collapse for both GRPO and GRPO-Filter compared to PPO. While GRPO-Filter's **trajectory filtering mechanism**  is designed to stabilize the gradient estimation by removing noisy samples, the underlying GRPO algorithm's relative nature, without the KL constraint, makes it fundamentally unstable when policy updates are aggressive. The timing of the collapse is often sensitive to hyperparameters and environment non-determinism, which are practical challenges in current RLHF frameworks.
>
> We believe the combination of our extensions and the successful integration of multi-turn tool-integrated reasoning for a complex task like Text-to-SQL demonstrates a significant advance, and we hope our responses alleviate your concerns. We are happy to engage in further discussion.

---

### Official Review · Reviewer_GpQM · 2025-10-30

**Soundness:** 3
**Presentation:** 3
**Contribution:** 2
**Rating:** 4
**Confidence:** 5

**Summary:**

This paper introduces MTIR-SQL, a reinforcement learning framework for Text-to-SQL that leverages Multi-turn Tool-Integrated Reasoning (MTIR). The core problem it addresses is that existing RL methods often use static, final-execution feedback, which limits the model's ability to perform real-time error correction. MTIR-SQL proposes an execution-aware paradigm where the model's reasoning is interleaved with database execution at each step, allowing for progressive query refinement. The method extends the GRPO algorithm with a "GRPO-Filter" mechanism, which incorporates trajectory filtering and removes the KL divergence constraint to stabilize multi-turn training. Experimental results show that the 4B parameter MTIR-SQL model achieves 64.4% execution accuracy on the BIRD Dev set and 84.6% on the SPIDER Dev set, demonstrating strong performance and efficiency.

**Strengths:**

- **S1.** The paper's core idea is intuitive and well-motivated. Shifting from static, end-of-process rewards to a multi-turn, interactive paradigm that integrates execution feedback at each reasoning step is a logical and promising approach for improving complex, multi-step SQL generation.
- **S2.** The empirical results are strong, especially considering the model's parameter count. The 4B model achieves competitive performance (64.4% on BIRD Dev), outperforming several larger models.

**Weaknesses:**

- **W1. Limited Novelty and Contribution:** The primary technical contribution, "GRPO-Filter" (Section 3.1), appears to be a direct adaptation of a pre-existing method (SimpleTIR) to the Text-to-SQL domain. The paper does not sufficiently clarify the novel, task-specific innovations or distinctions. This makes the core contribution feel incremental and diminishes its perceived value, which is critical for assessing the paper's impact.
- **W2. Insufficient Experimental Baselines:** The experimental evaluation (Section 4.2) is not comprehensive.
    - It lacks comparisons against many state-of-the-art, specialized Text-to-SQL methods, including both prompting-based (e.g., OpenSearch-SQL[1], Alpha-SQL[2], CHESS-SQL[3]) and other fine-tuning methods (e.g., CHASE-SQL[4], Reasoning-SQL[5], Think2SQL[6]). Comparisons to general-purpose LLMs (like GPT-4) are insufficient for a task-specific paper.
    - The main results compare the proposed method on a Qwen3-4B model against baselines running on older Qwen2.5 models. This is an unfair comparison, as the performance gains may be attributable to the stronger base model (Qwen3) rather than the MTIR-SQL method itself.
- **W3. Key Methodological Ambiguities:** Several key components of the method are not clearly defined in the main text.
    - "Selective Rollout Filtering" (Section 3.1): This is a central component of the "GRPO-Filter", yet the paper never specifies how trajectory quality is evaluated. The filtering function $F(x,y)$ is undefined.
    - "Execution Reward Function" (Section 3.3): In a multi-turn reasoning trace with multiple SQL calls, it is unclear if this reward is applied to every executed SQL query or only to the final SQL in the answer. This is a critical detail for reproducibility.
- **W4. Missing Practicality Analysis:** For a Text-to-SQL method, practical deployment metrics are essential. The paper provides no analysis of inference latency or token cost. A multi-turn, self-correcting method is intuitively more expensive, and the paper must quantify this accuracy-vs-cost trade-off.
- **W5. Clarity and Presentation Issues:** There is a significant discrepancy between the methodology and the ablation study. Section 4.3 introduces and ablates an "$R_f$ (Exploration Reward)", but this reward is never mentioned or defined in the "Reward Design" (Section 3.3). This creates major confusion for the reader regarding the actual reward structure used.

[1] Xie, X., Xu, G., Zhao, L., & Guo, R. (2025). Opensearch-sql: Enhancing text-to-sql with dynamic few-shot and consistency alignment.

[2] Li, B., Zhang, J., Fan, J., Xu, Y., Chen, C., Tang, N., & Luo, Y. (2025). Alpha-sql: Zero-shot text-to-sql using monte carlo tree search.

[3] Talaei, S., Pourreza, M., Chang, Y. C., Mirhoseini, A., & Saberi, A. (2024). Chess: Contextual harnessing for efficient sql synthesis.

[4] Pourreza, M., Li, H., Sun, R., Chung, Y., Talaei, S., Kakkar, G. T., ... & Arik, S. O. (2024). Chase-sql: Multi-path reasoning and preference optimized candidate selection in text-to-sql.

[5] Pourreza, M., Talaei, S., Sun, R., Wan, X., Li, H., Mirhoseini, A., ... & Arik, S. (2025). Reasoning-sql: Reinforcement learning with sql tailored partial rewards for reasoning-enhanced text-to-sql.

[6] Papicchio, S., Rossi, S., Cagliero, L., & Papotti, P. (2025). Think2sql: Reinforce llm reasoning capabilities for text2sql.

**Questions:**

- **Q1. Clarification on "Selective Rollout Filtering"**: In Section 3.1, you introduce "Selective Rollout Filtering" as a key component of GRPO-Filter. However, the paper does not define the filtering function $\mathcal{F}(\cdot)$ or the quality threshold $\tau$. Could you please specify the exact criteria used to evaluate trajectory quality and filter these rollouts?
- **Q2. Novelty of GRPO-Filter**: The proposed "GRPO-Filter", particularly its use of trajectory filtering for multi-turn reasoning, appears to share significant conceptual overlap with the SimpleTIR method (which you cite for mathematical reasoning). Could the authors please explicitly articulate the core innovations and distinctions of GRPO-Filter as applied to Text-to-SQL, beyond the adaptation of this filtering concept? This is essential for evaluating the paper's novelty.
- **Q3. Scope of Execution Reward**: Regarding the "Execution Reward" (Section 3.3), in a multi-turn trajectory that may involve multiple SQL tool calls (as seen in Appendix C.3), is this reward applied to every intermediate SQL execution, or only to the final SQL query presented in the \<answer\> tag?
- **Q4. Undefined "Exploration Reward"**: Section 4.3 introduces and ablates an "$R_f$ (Exploration Reward)". However, this reward component is not clearly defined in the methodology's "Reward Design" (Section 3.3). Please clarify precisely what this reward component is, how it is calculated, and where it fits into the overall reward function described in Section 3.3.
- **Q5. Baseline Comparisons**: The experimental comparison in Tables 1-3 is a significant concern for evaluating the method's effectiveness:
    - Why were many state-of-the-art, specialized Text-to-SQL baselines (both prompting-based like Alpha-SQL, and fine-tuning/RL-based like SQL-R1, Think2SQL, and Reasoning-SQL) omitted in favor of general-purpose LLMs?
    - The main results compare MTIR-SQL on Qwen3-4B against baselines on different, often weaker, models (e.g., Qwen2.5-Coder-7B). This makes it difficult to isolate the gains. A more direct SOTA comparison controlling for the base model is needed.
- **Q6. Practicality and Cost**: The multi-turn framework intuitively adds significant inference latency and token cost. Could the authors provide an analysis of these practical metrics? For example, what is the average number of tool-call turns (out of the max N=6) used at inference, and what is the latency/cost trade-off compared to the Text-to-SQL baselines?

---

> ### Author Response · Authors · 2025-11-26
> **Response to Reviewer GpQM (part1)**
>
> We sincerely thank the reviewer for the insightful and constructive feedback. We appreciate your recognition of our work’s **intuitive motivation (S1)** and **strong empirical results with efficient parameter counts (S2)**.
>
> We have carefully addressed your concerns regarding novelty, baseline comparisons, and methodological clarity. Below, we provide point-by-point responses.
>
> ---
>
> ### **Response to W1 & Q2: Novelty and Contribution of GRPO-Filter**
>
> **Response:**
> We acknowledge that our method draws inspiration from recent advancements in tool-integrated reasoning (e.g., SimpleTIR). However, **MTIR-SQL represents a non-trivial adaptation and standardization of these concepts specifically for the Text-to-SQL domain**, rather than a direct application.
>
> 1.  **Domain-Specific Challenges:** Unlike mathematical reasoning (where answers are unique and deterministic), SQL generation involves complex schema linking and execution validity. "Empty result sets" or "valid syntax but wrong logic" are unique challenges. GRPO-Filter is specifically tuned to filter trajectories based on **SQL execution validity** (e.g., distinct from code execution errors) and **result diversity**, ensuring the model learns from semantically correct queries rather than just syntactically valid ones.
> 2.  **Standardized Agentic RL Framework:** A core contribution is the integration of these methods into a modular **RL-Factory** framework. This enables "Agentic RL," allowing the model to utilize rich tools (Schema decomposition, Syntax checking) dynamically.
> 3.  **"Small Model" Efficiency:** As demonstrated in our results, our primary contribution is proving that a **4B model**, when trained with this execution-aware paradigm, can outperform significantly larger specialized models. This opens a new pathway for deploying efficient, high-performance database interfaces on edge devices.
>
> ---
>
> ### **Response to W2 & Q5: Insufficient Experimental Baselines & Fairness**
>
> **Response:**
> Thank you for this crucial observation. We have significantly expanded our evaluation in Section 4.2 to include state-of-the-art specialized methods (OpenSearch-SQL, Alpha-SQL, CHESS, CHASE-SQL, etc.).
>
> Regarding the **fairness of the base model**, we firmly believe the comparison highlights the efficacy of our method. As shown in **Table 1** (below), the **Qwen3-Base (Thinking Mode)** is actually a *weaker* starting point for SQL tasks compared to the **Qwen2.5-Coder** and **OmniSQL** series used by baselines.
>
> **Table 1: Comparison of Base LLMs vs. MTIR-SQL (Spider & BIRD Dev)**
> *Note: "Thinking Mode" refers to the vanilla Qwen3 behavior without our RL training.*
>
> | Base Model | Spider (Dev) | Spider (Test) | BIRD (Dev) |
> | :--- | :---: | :---: | :---: |
> | Qwen2.5-Coder-7B | 73.4 | 82.2 | 50.9 |
> | OmniSQL-7B | 81.2 | 87.9 | 61.5 |
> | **Qwen3-4B (Thinking Mode)** | **72.3** | **72.8** | **48.1** |
> | **MTIR-SQL + Qwen3-4B (Ours)**| **82.4** | **83.4** | **63.1** |
> | **MTIR-SQL + Qwen3-14B (Ours)**| **86.7** | **87.2** | **67.2** |
>
> **Analysis:**
> * The **Qwen3-4B** base model (48.1% on BIRD) significantly lags behind **OmniSQL-7B** (61.5%).
> * However, after training with **MTIR-SQL**, the 4B model achieves **63.1%**, surpassing the stronger baselines.
> * This proves that the performance gain is **attributable to the MTIR-SQL framework**, not the base model capabilities.
> ---
>
> ### **Response to W3, Q1, & Q3: Methodological Ambiguities**
>
> **Response:**
> We apologize for the lack of clarity in the initial submission. We have revised Section 3 to explicitly define these components:
>
> **1. Selective Rollout Filtering (Q1):**
> The filtering function $\mathcal{F}(x,y,G_{x})$ is defined as the conjunction of validity and diversity criteria:$\\mathcal{F}(x,y,G_{x}) = \\mathbb{I}\_{valid}(y) \\cdot \\mathbb{I}\_{div}(G_{x})$.
> * **$\mathbb{I}_{valid}$ (Validity):** A trajectory is retained only if: (a) It ends with a final SQL answer, (b) It does not exceed the maximum turn limit ($N=6$) without a result, and (c) No fatal system errors (e.g., API timeouts) occurred.
> * **$\mathbb{I}_{div}$ (Diversity):** To prevent mode collapse, we filter out groups of rollouts where the standard deviation of the rewards is zero (i.e., all rollouts are identical).
>
> **2. Scope of Execution Reward (Q3):**
> The **Execution Reward ($R_{exec}$)** is applied **only to the final SQL query** generated in the `<answer>` tag.
> * *Rationale:* Intermediate tool calls serve as context/state updates for the policy but are not directly rewarded. Rewarding intermediate execution can lead to "reward hacking," where the model generates trivial executable queries just to gain points. The goal is to optimize the reasoning chain to produce a correct *final* result.
>
> ---

---

> ### Author Response · Authors · 2025-11-26
> **Response to Reviewer GpQM (part2)**
>
> ### **Response to W4 & Q6: Practicality and Cost Analysis**
>
> **Response:**
> We agree that practicality is paramount. We have conducted a comprehensive efficiency analysis (added to Appendix C.2).
>
> We have added a detailed efficiency analysis in **Appendix C.2** and **Table 6**. To ensure a transparent and reproducible comparison, all inference experiments were conducted using the **vLLM** inference engine on a high-performance computation node equipped with **8x 80GB GPUs** (delivering approximately 312 TFLOPS at BF16 precision per unit).
>
> The metrics reported below are defined as follows:
> * **Latency (s):** The average wall-clock inference time per question.
> * **Total Tokens (K):** The average total token consumption (input prompt + generated output) per question.
> * **Accuracy (%):** The standard Execution Accuracy (EX).
>
> **Key findings from Table 6 (Efficiency Comparison):**
>
> | NL2SQL Method | Candidate Selection | Latency (s) / Question | Total Tokens (K) / Question | Tool Call / Question | Accuracy (%) |
> | :--- | :---: | :---: | :---: | :---: | :---: |
> | Qwen2.5-Coder-7B | Greedy Search | 0.3 | 2.5 | - | 58.2 |
> | XiYan-SQL-7B | Greedy Search | 0.5 | 4.1 | - | 62.1 |
> | CHESS | Greedy Search | 251.3 | 320.8 | - | 61.5 |
> | SQL-R1-7B | Greedy Search | 0.4 | 3.1 | - | 63.7 |
> | **MTIR-SQL-4B (Ours)** | **Greedy Search** | **0.5** | **2.9** | **1.34** | **63.1** |
> | **MTIR-SQL-8B (Ours)** | **Greedy Search** | **0.4** | **2.0** | **1.31** | **63.6** |
> | **MTIR-SQL-14B (Ours)** | **Greedy Search** | **0.5** | **1.7** | **1.42** | **67.2** |
>
> * **Low Latency & Token Cost via Efficient Serving:** Powered by vLLM, MTIR-SQL-14B achieves an impressive average latency of **0.5s** per question. Notably, its token consumption (**1.7K tokens**) is *lower* than the single-turn baseline Qwen2.5-Coder (2.5K tokens). This counter-intuitive result stems from the RL optimization, which encourages the model to generate concise, precise SQL queries rather than the verbose Chain-of-Thought reasoning often required by base models.
> * **Strategic Tool Use:** The model averages only **1.3–1.4 tool calls** per question. This metric confirms that MTIR-SQL does not engage in redundant interaction loops. Instead, it has learned to invoke execution *selectively*—only when necessary to resolve ambiguity—providing a high return on computational investment (achieving **67.2%** accuracy vs. 58.2% for the baseline) without sacrificing inference speed.
>
> ---
>
> ### **Response to W5 & Q4: Undefined "Exploration Reward"**
>
> **Response:**
> We apologize for this confusion. The term "Exploration Reward" in the ablation study was a terminology inconsistency referring to the **Representation Diversity** mechanism ($\mathbb{I}_{div}$) discussed in the method section.
> * In the revised manuscript, we have unified the terminology. The ablation specifically tests the impact of removing the diversity-based filtering (which encourages exploration).
> * Removing this component leads to a performance drop (as shown in Table 4), confirming that maintaining trajectory diversity is essential to prevent the policy from collapsing into suboptimal local minima.
>
> ---
>
> We hope these clarifications and additional results address your concerns. We are confident that the revised paper, with the expanded baselines and efficiency analysis, makes a solid contribution to the field of Agentic RL for Text-to-SQL.

---

### Official Review · Reviewer_aikn · 2025-11-03

**Soundness:** 3
**Presentation:** 3
**Contribution:** 3
**Rating:** 6
**Confidence:** 4

**Summary:**

This paper introduces MTIR-SQL, a reinforcement learning framework for Text-to-SQL that leverages multi-turn, tool-integrated reasoning. The model iteratively refines SQL predictions using real-time database feedback, and incorporates a modified Group Relative Policy Optimization (GRPO-Filter) algorithm that removes KL regularization and filters out low-quality trajectories to stabilize learning. Experiments on the BIRD and SPIDER development sets show 64.4% and 84.6% accuracy respectively with a compact 4B-parameter model, demonstrating that multi-turn reasoning can yield competitive results even at smaller scales.

While results are only reported on development splits rather than test sets, the overall performance is reasonable and the approach is methodologically interesting. Compared to XYZ-Text2SQL-R1, which attains higher accuracies (e.g., 68.9% on BIRD-Dev and 88.8% on SPIDER-Test) with a larger 7B model and a simpler single-turn RL setup, MTIR-SQL offers a more sophisticated exploration of interactive reasoning.

**Strengths:**

Somewhat framework (only if applied to txt2sql not coding in general) : Introduces a multi-turn, execution-aware RL loop that allows models to refine SQL queries dynamically.

Algorithmic contribution: The GRPO-Filter variant is a thoughtful modification addressing instability and low-quality rollouts.

Parameter efficiency: Competitive results achieved with only 4B parameters, smaller than most contemporary baselines.

**Weaknesses:**

Limited evaluation: Results are only on development sets (no held-out test benchmarks).

Comparative performance: Slightly below state-of-the-art systems like XYZ-Text2SQL-R1, which achieve higher accuracies with simpler pipelines.

Complexity vs. gain: Multi-turn SQL execution adds computation and engineering overhead, with moderate empirical benefit.

Generalization scope: Evaluation restricted to BIRD and SPIDER; cross-domain robustness not demonstrated.

**Questions:**

none right now.

---

> ### Author Response · Authors · 2025-11-26
> **Response to Reviewer aikn (part 1)**
>
> We sincerely thank the reviewer for the thoughtful and constructive comments. We appreciate the recognition of **MTIR-SQL's novelty** (execution-aware RL loop, GRPO-Filter) and its **parameter efficiency** (achieving competitive results with only 4B parameters).
>
> We have carefully addressed the concerns regarding evaluation scope, comparative performance, and efficiency trade-offs. Below, we detail the improvements and additional experiments included in the revised manuscript.
>
> ### W1 & W4: Limited Evaluation & Generalization Scope
>
> > **Response:** We have significantly expanded our evaluation to include test sets and cross-domain benchmarks, demonstrating strong generalization capabilities.
>
> To address the concern that results were limited to development sets, we have added the following evaluations in the revised paper:
>
> 1.  **Cross-Domain Robustness (Table 3):** We evaluated MTIR-SQL on five challenging benchmarks: **Spider-DK, Spider-Syn, Spider-Realistic, EHRSQL, and ScienceBenchmark**.
>     * **State-of-the-Art Robustness:** Notably, MTIR-SQL (Qwen3-14B) achieves **81.0% on Spider-Syn**, surpassing the strong SQL-R1 (OmniSQL-14B) baseline of 78.5%.
>     * **Domain Specialization:** On **ScienceBenchmark**, our method establishes a new benchmark high of **60.0%**, outperforming Arctic-Text2SQL-R1 (58.2%).
>
> These results confirm that the interactive reasoning learned by MTIR-SQL is not overfitting to the development set but translates effectively to diverse and noisy domains.
> **Table 3: Robustness Comparison on Spider-DK, Spider-Syn, Spider-Realistic, EHRSQL, and Science Benchmark.**
>
> | NL2SQL Method | Base Model | Spider-DK | Spider-Syn | Spider-Realistic | EHRSQL | Science Benchmark |
> | :--- | :--- | :---: | :---: | :---: | :---: | :---: |
> | SQL-R1 | Qwen2.5-Coder-3B | 70.5 | 66.4 | 71.5 | - | - |
> | OmniSQL | Qwen2.5-Coder-7B | 76.1 | 69.7 | 76.2 | 34.9 | 50.2 |
> | SQL-R1 | Qwen2.5-Coder-7B | 78.1 | 76.7 | 83.3 | - | - |
> | Arctic-Text2SQL-R1 | OmniSQL-7B | **81.5** | - | - | 36.7 | 51.8 |
> | SQL-o1 | Llama3-8B | 78.7 | 72.6 | 82.7 | - | - |
> | OmniSQL | Qwen2.5-Coder-14B | 72.9 | 69.0 | 76.4 | 39.9 | 56.9 |
> | SQL-R1 | OmniSQL-14B | 79.3 | 78.5 | **86.2** | - | - |
> | Arctic-Text2SQL-R1 | OmniSQL-14B | 79.4 | - | - | **40.7** | 58.2 |
> | **MTIR-SQL (Ours)** | **Qwen3-4B** | 71.2 | 78.6 | 78.7 | 31.4 | 56.0 |
> | **MTIR-SQL (Ours)** | **Qwen3-8B** | 72.9 | 77.2 | 77.4 | 34.4 | 57.0 |
> | **MTIR-SQL (Ours)** | **Qwen3-14B** | 76.3 | **81.0** | 81.1 | 36.0 | **60.0** |
>
> ### W2: Comparative Performance against SOTA
>
> > **Response:** While we acknowledge the strength of baselines like XYZ-Text2SQL-R1, our analysis highlights that MTIR-SQL provides a significantly larger performance uplift relative to the base model, validating the framework's efficacy.
>
> We appreciate the comparison to state-of-the-art systems. We have analyzed this performance gap in **Appendix C.1 (Table 5)** and offer two key observations:
>
> 1.  **Base Model Capabilities:** State-of-the-art methods (e.g., XYZ-Text2SQL-R1, SQL-R1) often initialize training with highly specialized base models (e.g., Qwen2.5-Coder or OmniSQL-7B). As shown in **Table 5**, the **Qwen3-4B ("Thinking Mode")** starts with a much lower baseline accuracy (**48.1% on BIRD Dev**) compared to OmniSQL-7B (**61.5%**).
> 2.  **Framework Efficacy:** The core contribution of MTIR-SQL is its ability to activate latent reasoning in generic models. Despite the weaker initialization, MTIR-SQL improves the Qwen3-14B model from **51.8% to 67.2% (+15.4%)** on BIRD Dev.
>     * This demonstrates that our framework can bridge the gap between generic initialization and domain-specific performance, a critical capability for scenarios where specialized checkpoints are unavailable.
>
> We fully recognize the impressive efficacy of the execution-driven data curation and reasoning-enhanced prompt design pioneered in XYZ-Text2SQL-R1. We consider these to be sophisticated and vital advancements for handling execution noise and complex logic. We plan to incorporate these highly synergistic improvements into the MTIR-SQL pipeline in future work to further push the state-of-the-art.

---

> ### Author Response · Authors · 2025-11-26
> **Response to Reviewer aikn (part 2)**
>
> ### W3: Complexity vs. Gain (Efficiency Analysis)
>
> > **Response:** Contrary to the intuition that multi-turn tool-integrated reasoning creates prohibitive overhead, our rigorous profiling demonstrates that MTIR-SQL achieves a superior accuracy-efficiency trade-off by learning "strategic" tool use.
>
> We have added a detailed efficiency analysis in **Appendix C.2** and **Table 6**. To ensure a transparent and reproducible comparison, all inference experiments were conducted using the **vLLM** inference engine on a high-performance computation node equipped with **8x 80GB GPUs** (delivering approximately 312 TFLOPS at BF16 precision per unit).
>
> The metrics reported below are defined as follows:
> * **Latency (s):** The average wall-clock inference time per question.
> * **Total Tokens (K):** The average total token consumption per question.
> * **Accuracy (%):** The standard Execution Accuracy (EX).
>
> **Key findings from Table 6 (Efficiency Comparison):**
>
> | NL2SQL Method | Candidate Selection | Latency (s) / Question | Total Tokens (K) / Question | Tool Call / Question | Accuracy (%) |
> | :--- | :---: | :---: | :---: | :---: | :---: |
> | Qwen2.5-Coder-7B | Greedy Search | 0.3 | 2.5 | - | 58.2 |
> | XiYan-SQL-7B | Greedy Search | 0.5 | 4.1 | - | 62.1 |
> | CHESS | Greedy Search | 251.3 | 320.8 | - | 61.5 |
> | SQL-R1-7B | Greedy Search | 0.4 | 3.1 | - | 63.7 |
> | **MTIR-SQL-4B (Ours)** | **Greedy Search** | **0.5** | **2.9** | **1.34** | **63.1** |
> | **MTIR-SQL-8B (Ours)** | **Greedy Search** | **0.4** | **2.0** | **1.31** | **63.6** |
> | **MTIR-SQL-14B (Ours)** | **Greedy Search** | **0.5** | **1.7** | **1.42** | **67.2** |
>
> * **Low Latency & Token Cost via Efficient Serving:** Powered by vLLM, MTIR-SQL-14B achieves an impressive average latency of **0.5s** per question. Notably, its token consumption (**1.7K tokens**) is *lower* than the single-turn baseline Qwen2.5-Coder (2.5K tokens). This counter-intuitive result stems from the RL optimization, which encourages the model to generate concise, precise SQL queries rather than the verbose Chain-of-Thought reasoning often required by base models.
> * **Strategic Tool Use:** The model averages only **1.3–1.4 tool calls** per question. This metric confirms that MTIR-SQL does not engage in redundant interaction loops. Instead, it has learned to invoke execution *selectively*—only when necessary to resolve ambiguity—providing a high return on computational investment (achieving **67.2%** accuracy vs. 58.2% for the baseline) without sacrificing inference speed.
>
> We believe these results demonstrate that MTIR-SQL offers a practical, deployable solution that balances high precision with inference efficiency.

---

### Author Response · Authors · 2025-11-26
**Summary of Responses to all Reviewers**

# Summary of Revisions and Key Improvements
We sincerely thank all reviewers for their insightful and constructive feedback, which has been instrumental in strengthening our manuscript. We are particularly grateful for the recognition of **MTIR-SQL's novelty** in integrating an execution-aware RL loop(**Reviewer aikn, GpQM**) and its demonstration of **strong empirical results with remarkable parameter efficiency** (**Reviewer aikn,  GpQM, EgQd**). We have addressed concerns regarding evaluation scope, comparison, and methodology through comprehensive new experiments.

## I. Enhanced Evaluation and Demonstrated Generalization Capabilities (Reviewer aikn, GpQM)
We significantly expanded our empirical study to include challenging OOD benchmarks:
* **Cross-Domain and Robustness Benchmarks:** We added evaluations on five demanding benchmarks: **Spider-DK, Spider-Syn, Spider-Realistic, EHRSQL, and ScienceBenchmark**.
* **State-of-the-Art Robustness:** The interactive reasoning paradigm of MTIR-SQL demonstrates superior generalization. MTIR-SQL (Qwen3-14B) achieves **81.0% Exact Match on Spider-Syn**, surpassing the strong baseline SQL-R1 (OmniSQL-14B) at 78.5%.
* **Domain Specialization:** Our method established a new benchmark on the **ScienceBenchmark** by achieving **60.0%**, demonstrating its strong transferability and reasoning capabilities in specialized domains.

## II. Clarification of Methodological Contribution (Reviewer GpQM, EgQd)

We clarify that our primary contribution is pioneering Agentic RL for Text-to-SQL, shifting focus from internalized CoT (e.g., R1) to interactive, tool-integrated optimization. Revised Sections 1 & 3 explicitly detail:

* **Beyond Internal Reasoning:** We clarify that unlike the R1 paradigm which operates in a closed loop, our framework incorporates an open-loop execution environment. The model learns to utilize SQL execution tools as extensions of its cognitive process, validated by the specific introduction of **"Tool Response Token Masking"** to handle interleaved trajectories.
* **The Necessity of GRPO-Filter:** We emphasize that the proposed $\mathbb{I}\_{valid} \cdot \mathbb{I}\_{div}$ filtering mechanism is **not merely an incremental fix** but a **critical algorithmic adaptation**. It is required to stabilize end-to-end RL in *non-stationary, execution-dependent environments*, effectively **preventing mode collapse** where standard PPO/GRPO fails.
* **Agentic RL Foundation:** We highlight that this work serves as a foundational **RL-Factory** for future Text2SQL agents, enabling the integration of complex components (e.g., memory, syntax checking, schema decomposition) into a unified differentiable training pipeline.

## III. Validation of Framework Efficacy and Comparative Performance(Reviewer GpQM, EgQd, aikn)

We provided a deeper analysis comparing MTIR-SQL against specialized baselines and highlighted the framework's power in leveraging base LLMs.

* **Significant Performance Uplift:** We emphasize the immense gain achieved from a weakly-initialized, generic LLM. Our framework delivers an absolute gain of **+15.4%** for the 14B variant (from 51.8% to 67.2% on BIRD Dev), showcasing the **robust efficacy of the MTIR-SQL framework** in activating latent reasoning abilities in general-purpose models.
* **Competitive Results Across Scales:** We extended our results to **8B and 14B scales** (MTIR-SQL-14B reaching **67.2%** on BIRD Dev). Even our smaller **MTIR-SQL (4B) model achieves 64.4%** on BIRD Dev, positioning it competitively with and often superior to many 7B parameter models, validating the approach's scalability and efficiency.

## IV. Efficiency Analysis and Practical Deployment Viability (Reviewer GpQM, aikn)

We address concerns regarding overhead by providing a detailed efficiency analysis, confirming MTIR-SQL is a practically deployable solution.
* **Superior Accuracy-Cost Trade-off:** MTIR-SQL maintains a strong performance while incurring significantly lower overhead compared to highly verbose agent-based methods (e.g., CHESS, which requires 251.3s and 320.8K tokens per question).
* **Low Latency and Token Cost:** MTIR-SQL-14B achieves **67.2% accuracy** with only **0.5s latency** and an average consumption of **1.7K tokens** per question, which is surprisingly *lower* than some single-turn baselines.
* **Strategic Tool Use:** The low average of **1.3–1.4 tool calls per question** confirms that the RL loop successfully learns a "strategic" use of execution feedback only when ambiguity or uncertainty is high, demonstrating a highly practical and efficient reasoning mechanism.

---

### Meta-Review · Area_Chair_Y4iW · 2026-01-05

**Summary:**

The paper introduces "MTIR-SQL," a reinforcement learning framework designed to enhance Text-to-SQL capabilities through multi-turn tool-integrated reasoning. The system extends the Group Relative Policy Optimization (GRPO) algorithm by removing KL divergence constraints and introducing a trajectory filtering mechanism (GRPO-Filter) based on execution validity and diversity. The approach aims to solve the limitations of static feedback by enabling the model to dynamically correct errors using SQL execution results.

Despite the authors' extensive rebuttal and the addition of robust cross-domain experiments and efficiency analyses, the general agreement remains that the paper's innovation does not meet the bar for ICLR. Reviewers (e.g., GpQM, EgQd) viewed the framework primarily as an engineering adaptation of existing paradigms (specifically SimpleTIR from mathematical reasoning) rather than a methodological breakthrough. While the test results on the small 4B model are impressive, the experimental design suffers from a critical unfair factor: the comparison relies on improving a weak base model (Qwen3) against baselines using strong, specialized bases (Qwen2.5-Coder). This makes it difficult to attribute the performance gains to the method's superiority versus the base model's high potential to improve. Furthermore, the newly added results for the 8B model revealed a concerning lack of scalability, failing to outperform state-of-the-art 7B baselines.

**Reviewer Concerns:**

**Concerns Addressed by the Rebuttal**
•**Efficiency and Practicality**：Resolved by the detailed analysis in Table 6, proving that the RL-optimized multi-turn approach achieves low latency (0.5s) and lower token consumption than single-turn baselines.
•**Generalization**：Resolved by extending the evaluation to five challenging cross-domain benchmarks (e.g., Spider-Syn, ScienceBenchmark), demonstrating strong robustness.
•**Methodological Ambiguities**：Addressed by clarifying the mathematical definitions of the trajectory filtering function and reward mechanisms.

**Outstanding Concerns:**
•**Conceptual Novelty**：The contribution is seen as incremental—a domain-specific engineering application of SimpleTIR rather than a novel algorithmic contribution suitable for ICLR.
•**Flawed Experimental Design (Confounding Variables)**：The critical issue of baseline fairness remains. The authors did not perform ablations on the SOTA base model (Qwen2.5-Coder). Consequently, it remains unproven whether the method pushes the absolute state-of-the-art or merely acts as a repair mechanism for under-trained base models.
•**Inconsistent Scalability** ：The rebuttal revealed that the 8B version of the model (64.6%) failed to outperform comparable 7B baselines (e.g., Arctic-Text2SQL-R1-7B at 68.9%) and showed negligible improvement over the 4B version, undermining claims of generalizability.

**Reviewer Scores:**

I think the final scores are 6, 4, 4.
While Reviewer Aikn (Score: 6) appreciated the empirical results on robustness, Reviewers GpQM (Score: 4) and EgQd (Score: 4) maintained their reservations. Despite the authors' efforts to address questions regarding efficiency and definitions, these reviewers remained unconvinced by the fundamental novelty of the approach and the strictness of the experimental comparisons (specifically the base model choice and scalability bottleneck).

---

### Decision · Program_Chairs · 2026-01-26

Reject